# Epigenetic regulation by polycomb repressive complex 1 promotes cerebral cavernous malformations

Van-Cuong Pham [ID] [1,16], Claudia Jasmin Rödel [ID] [1,16], Mariaelena Valentino [ID] [2], Matteo Malinverno[2], Alessio Paolini [ID] [1], Juliane Münch [ID] [1], Candice Pasquier [ID] [3], Favour C Onyeogaziri [ID] [4], Bojana Lazovic[5,6,7], Romuald Girard[8], Janne Koskimäki [ID] [8], Melina Hußmann [ID] [9], Benjamin Keith[5], Daniel Jachimowicz[10], Franziska Kohl[5,11], Astrid Hagelkruys [ID] [12], Josef M Penninger [ID] [12,13], Stefan Schulte-Merker [ID] [9], Issam A Awad[8], Ryan Hicks[5,14], Peetra U Magnusson [ID] [4], Eva Faurobert [ID] [3], Massimiliano Pagani [ID] [2,15 ✉] & Salim Abdelilah-Seyfried [ID] [1✉]

## Abstract

Cerebral cavernous malformations (CCMs) are anomalies of the cerebral vasculature. Loss of the CCM proteins CCM1/KRIT1, CCM2, or CCM3/PDCD10 trigger a MAPK-Krüppel-like factor 2 (KLF2) signaling cascade, which induces a pathophysiological pattern of gene expression. The downstream target genes that are activated by KLF2 are mostly unknown. Here we show that Chromobox Protein Homolog 7 (CBX7), component of the Polycomb Repressive Complex 1, contributes to pathophysiological KLF2 signaling during zebrafish cardiovascular development. *CBX7/ cbx7a* mRNA is strongly upregulated in lesions of CCM patients, and in human, mouse, and zebrafish CCM-deficient endothelial cells. The silencing or pharmacological inhibition of CBX7/Cbx7a suppresses pathological CCM phenotypes in *ccm2* zebrafish, CCM2-deficient HUVECs, and in a pre-clinical murine *CCM3* disease model. Whole-transcriptome datasets from zebrafish cardiovascular tissues and human endothelial cells reveal a role of CBX7/ Cbx7a in the activation of KLF2 target genes including *TEK*, *ANGPT1*, *WNT9*, and endoMT-associated genes. Our findings uncover an intricate interplay in the regulation of Klf2-dependent biomechanical signaling by CBX7 in CCM. This work also provides insights for therapeutic strategies in the pathogenesis of CCM.

**Keywords** CBX7; Cerebral Cavernous Malformation; endoMT; KLF2; WNT9

**Subject Categories** Cardiovascular System; Vascular Biology & Angiogenesis

## Introduction

Cerebral cavernous malformations (CCMs) occur as sporadic or hereditary familial forms and are characterized by hemorrhages mainly within capillary-venous vessel beds in the brain. These lesions can occur throughout a lifetime with severe neurological or even fatal consequences (Abdelilah-Seyfried et al, 2020; Snellings et al, 2021). Currently, therapeutic approaches mainly involve surgical methods, which are limited when CCM lesions develop in in-operable regions of the brain or spinal cord. Hence, pharmacological treatment options are urgently needed either to prevent the onset of lesion formation or to trigger the remission of acute lesions. Familial forms of CCM have been linked to loss-of-function mutations in genes encoding the proteins Krev interaction trapped protein 1 (KRIT1)/CCM1 (Laberge-le Couteulx et al, 1999), Malcavernin/CCM2 (Liquori et al, 2003), or programmed cell death 10 (PDCD10)/CCM3 (Bergametti et al, 2005). The loss of either one of these proteins results in similar disease phenotypes,

[1]Institute of Biochemistry and Biology, Potsdam University, D-14476 Potsdam, Germany. [2]IFOM ETS - The AIRC Institute of Molecular Oncology, Milan 20139, Italy. [3]University Grenoble Alpes UGA, CNRS 5309 INSERM 1209, Grenoble, France. [4]Department of Immunology, Genetics and Pathology, Uppsala University, 75185 Uppsala, Sweden. [5]Translational Genomics, Discovery Sciences, BioPharmaceuticals R&D, AstraZeneca, 43183 Mölndal, Gothenburg, Sweden. [6]Oulu Center for Cell-Matrix Research, Biocenter Oulu and Faculty of Biochemistry and Molecular Medicine, University of Oulu, 90220 Oulu, Finland. [7]BioPharmaceuticals R&D Cell Therapy, Research and Early Development, Cardiovascular, Renal and Metabolism (CVRM), BioPharmaceuticals R&D, AstraZeneca, 43183 Mölndal, Gothenburg, Sweden. [8]Department of Neurological Surgery, University of Chicago Medicine and Biological Sciences, Chicago, IL 60637, USA. [9]Institute for Cardiovascular Organogenesis and Regeneration, Medical Faculty, WU Münster, D-48149 Münster, Germany. [10]Data Sciences and Quantitative Biology, Discovery Sciences, R&D, AstraZeneca, 43183 Mölndal, Gothenburg, Sweden. [11]Department of Medical Biochemistry and Biophysics, Karolinska Institutet, 17165 Solna, Stockholm, Sweden. [12]IMBA, Institute of Molecular Biotechnology of the Austrian Academy of Sciences, 1030 Vienna, Austria. [13]Helmholtz-Centre for Infection Research, D-38124 Braunschweig, Germany. [14]School of Cardiovascular and Metabolic Medicine and Sciences, King's College London, WC2R 2LS London, United Kingdom. [15]Department of Medical Biotechnology and Translational Medicine, Università degli Studi di Milano, 20133 Milan, Italy. [16]These authors contributed equally: Van-Cuong Pham, Claudia Jasmin Rödel. ✉E-mail: massimiliano.pagani@ifom.eu; salim.seyfried@uni-potsdam.de

which are caused by enhancing signaling via MAP3K3/ERK5 and Krüppel-like transcription factors KLF2/4 (Zhou et al, 2015, 2016; Uhlik et al, 2003; Cuttano et al, 2016; Fisher et al, 2015; Chapman et al, 2019). This is a key regulatory pathway of biomechanical responses within endothelial cells (Dekker et al, 2005, 2006; Egorova et al, 2011). Functional studies have shown that CCM-associated cardiovascular defects can be suppressed when CCM-deficient mice undergo a loss of Klf2/4 (Zhou et al, 2015, 2016; Cuttano et al, 2016) or zebrafish *ccm* mutants lack Klf2a/b (Renz et al, 2015). Klf2 can trigger an endothelial-to-mesenchymal transition (endMT) (Dejana and Lampugnani, 2018), which is thought to contribute to the formation of CCM lesions because endothelial cells acquire mesenchymal properties (Maddaluno et al, 2013; Bravi et al, 2016; Dejana and Lampugnani, 2018). These discoveries have established KLF2 signaling as a key player in CCM, but its downstream targets in the etiopathology of the disease are largely unknown.

Several studies have demonstrated that blood flow affects the severity and occurrence of pathological phenotypes in CCM models. Zebrafish completely lacking Krit1 protein exhibited severe defects in the heart, preventing blood flow (Mably et al, 2006; Rödel et al, 2019; Renz et al, 2015). In this physiological condition without blood flow, even major blood vessels developed CCM phenotypes (Mably et al, 2006; Hogan et al, 2008; Renz et al, 2015; Rödel et al, 2019). Strikingly, expressing Krit1 specifically in the heart restored blood flow and suppressed the pathological phenotypes in these major blood vessels (Rödel et al, 2019). This suggests that high levels of blood flow exert protective effects, in contrast to regions of low or no blood flow where CCM lesions occur. Apparently, Klf2a has activities that switch from vasoprotective targets under conditions of strong blood flow to pathological targets in the latter conditions. Similarly, human endothelial cells lacking either KRIT1 or CCM2 and exposed to high levels of fluid shear stress had transcriptional profiles reminiscent of wild-type. Under low fluid shear stress conditions, however, these endothelial cells exhibited a transcriptional response that was characteristic of CCM signaling (Li et al, 2019). These studies suggest that pathological signaling in CCM occurs in the absence of strong fluid shear stress. They also point at a molecular mechanism that can direct Klf2 towards different target genes, in a manner that is contingent upon the strength of blood flow.

Proteins of the polycomb group are central regulators of chromatin states during embryogenesis. They control epigenetic changes to the genome that affect diverse biological processes, including differentiation programs, cell cycle progression, stem cell states, and endothelial-to-mesenchymal transition (Piunti and Shilatifard, 2021; Wu and Yang, 2011; Yang et al, 2010). Polycomb group proteins assemble into two separate complexes with different histone-modifying activities. The polycomb repressive complex 2 (PRC2) comprises the methyltransferases Enhancer of Zeste Homologs 1 and 2 (EZH1/2), which tri-methylate Histone 3 on lysine 27 residues (H3K27me3) (Czermin et al, 2002; Müller et al, 2002; Kuzmichev et al, 2002; Cao et al, 2002). This is a repressive biochemical modification, which is recognized by Chromobox protein homologs (CBX) of the PRC1, which direct this second multi-protein complex into the proximity of PRC2. The main enzymes of PRC1 are the E3 ubiquitin ligases Ring Finger Proteins 1 A/B (RING1A/B), which mono-ubiquitinate Histone H2A on lysine 119 (H2AK119ub) (de Napoles et al, 2004; Wang et al,

2004a). These epigenetic marks result in chromatin compaction and transcriptional silencing of genomic sites (Wang et al, 2004a, 2004b).

CBX7 is a component of the PRC1 and an important epigenetic regulator of developmental processes including epithelial-to-mesenchymal transition (EMT) (Klauke et al, 2013; Li et al, 2020). This is a process related to the endothelial-to-mesenchymal transition (endoMT), which has been implicated in CCM (Maddaluno et al, 2013). A gain-of-function activation of polycomb group proteins has been observed in various cancers (Parreno et al, 2022). Cases in which CBX7 is strongly upregulated include gastric cancer (Zhang et al, 2010), germinal center-derived follicular lymphomas (Scott et al, 2007), and acute myeloid leukemia, where it promotes proliferation, suppresses differentiation, and contributes to the self-renewal of stem cells (Jung et al, 2019).

Here, we show that a loss of Cbx7a/CBX7 suppresses CCM phenotypes in Ccm-deficient zebrafish and human umbilical vein endothelial cells (HUVECs). In zebrafish, Cbx7a expression is suppressed by blood flow but activated by Klf2a. In turn, Cbx7a directs Klf2a towards pathological target genes. To identify such pathological downstream targets, we used a transgenic model for the endothelial-specific over-expression of Klf2a. Quantifying the expression levels of candidate genes and assaying the effects of co-silencing Cbx7a implicated the Tie2 receptor (encoded by *tek*) and Wnt9 signaling pathways, which we functionally characterized in zebrafish CCM models. Our findings uncover a critical role of PRC-dependent epigenetic regulation in the pathogenesis of CCM. We demonstrate that the pharmacological inhibition of Cbx7a prevents cardiovascular phenotypes in a zebrafish CCM2 model and in a pre-clinical murine disease model of CCM3. This identifies CBX7 as a particularly interesting target for the pharmacological therapy of CCM.

# Results

## *cbx7a* is upregulated in CCM

To identify molecular mechanisms involved in the transcriptional regulation of pathological target genes in CCM, we assessed published microarray and RNA-seq datasets obtained from zebrafish *ccm2^{m201}* mutants hearts and mouse conditional endothelial-specific *Krit1* knockouts (Renz et al, 2015; Koskimäki et al, 2019). One of the most highly upregulated genes in zebrafish *ccm2^{m201}* mutants encodes the PRC1 component Cbx7a (Fig. 1A,B) (Renz et al, 2015). The mRNA expression of *cbx7a* is also elevated in zebrafish *krit1^{ty219c}* mutant hearts, expression of its homolog, *Cbx7* is elevated in conditional, endothelial-specific *Krit1* knockout mice (Koskimäki et al, 2019), and *CBX7* is upregulated in human iPSC-derived endothelial cells (ECs) (Fig. 1B). We next examined the expression of other components of the PRC1 in published whole-transcriptome datasets from zebrafish *ccm2^{m201}* mutants (Renz et al, 2015). Compared to the wild-type, a few genes encoding PRC1 proteins were significantly changed (Dataset EV1). This encouraged us to further characterize the role of Cbx7a and PRC1 in zebrafish models of CCM.

To elucidate the spatial expression pattern of *cbx7a* mRNA within the zebrafish cardiovascular system, we performed whole-mount in situ hybridizations. We found that *cbx7a* was expressed at

high levels throughout the entire endocardium in *ccm2^m201* and *krit1^ty219c* mutants (Figs. 1D,E and EV1B,D), compared to low levels of expression in wild-type (Figs. 1C and EV1A,C). Additionally, we quantified levels of *cbx7a* expression in different vascular beds using qRT-PCR on purified hearts, head and tail preparations (Fig. 1F). This revealed a strong upregulation of *cbx7a* mRNA levels in both, cardiac tissue and in tail regions, suggesting it is widely upregulated in the entire cardiovascular system of zebrafish *ccm* mutants.

To explore whether CBX7 was upregulated in CCM patients as well, we conducted immunohistological stainings of CBX7 in primary lesion material from familial cases of CCM. Immunohistochemistry with an antibody against CBX7 revealed strong expression in ECs from brain lesions and surrounding neural tissues of patients with familial CCM1 and CCM2 (Fig. 1G–J). Co-stained material from healthy control tissues revealed lower levels of CBX7 in ECs (Fig. 1K).

## Loss of Cbx7a suppresses cardiovascular phenotypes in zebrafish *ccm* mutants

Next, we aimed at a functional characterization of Cbx7a in the context of zebrafish CCM models. We generated *cbx7a* loss-of-function mutants using CRISPR/Cas9-mediated genome editing. This was achieved by targeting and deleting 510 bp of the presumptive promoter region of the gene locus (Fig. 2A) because such deletion mutants are likely to evade genetic compensation mechanisms (Rossi et al, 2015; El-Brolosy et al, 2019). Mutants of the promoter-less allele *cbx7a^pbb62* completely lacked wild-type *cbx7a* mRNA (Fig. 2B), but did not show gross morphological changes as compared to their wild-type siblings (Appendix Fig. S1A,B). We also found that the expression levels of the paralogous gene *cbx7b* were not altered in *cbx7a^pbb62* or *ccm2^m201* single mutants, or in *ccm2^m201;cbx7a^pbb62* double mutants (Appendix Fig. S1C). However, a more careful characterization of *cbx7a^pbb62* mutants revealed that endocardial cell numbers were slightly reduced when compared to wild-type (Appendix Fig. S1D).

To functionally test whether Cbx7a contributes to *ccm* mutant cardiovascular phenotypes, we generated *ccm2^m201;cbx7a^pbb62* double mutant embryos and characterized cardiovascular development. At 56 h post fertilization (hpf), the cardiac chambers are separated by the atrioventricular canal (AVC), a narrowing between both chambers (Fig. 2C, arrowhead). The endocardial cells of the AVC will form the future cardiac valves and are marked by activated leukocyte cell adhesion molecule (ALCAM) expression (Fig. 2C'). In comparison to wild-type, *ccm2^m201* mutants exhibit significantly enlarged hearts (Fig. 2D), a lack of endocardial cushion cells at the AVC (Fig. 2D'), and lack of blood flow (Fig. EV2A). The mutant hearts also lack a thickened layer of cardiac jelly (Fig. 2D') that separates the endocardium from the myocardium in wild-type (Fig. 2C'; asterisk). Abolishing the function of Cbx7a in *ccm2^m201;cbx7a^pbb62* double mutants normalized heart morphology and function (Fig. 2E), which restored blood flow in a subset of these double mutants (Fig. EV2A). In *ccm2^m201;cbx7a^pbb62* double mutants, the morphological narrowing of the heart tube at the AVC was normalized to a wild-type morphology, and ALCAM-positive endocardial cushion cells were present (Fig. 2E,E'). Similarly, endocardial cell numbers, which are increased in *ccm2^m201* mutants, were normalized to wild-type levels (Fig. 2F). Also, other CCM-related defects in different vascular beds including an

overproliferation of the lateral dorsal aorta and an abnormal network formation of the caudal vein plexus were restored upon a loss of Cbx7a in *ccm2^m201* mutants (Fig. EV2B–M). The suppression of the CCM-associated phenotypes could not be explained by normalized *klf2a* mRNA expression levels, which, along with the expression of its orthologue *klf2b*, remained elevated in phenotypically rescued *ccm2^m201;cbx7a^pbb62* double mutants as assessed by qRT-PCR analyses of whole embryos (Fig. 2G). A similar rescue effect was observed in *krit1^ty219c* mutants when knocking down *cbx7a* using an antisense morpholino oligo. This restored heart morphology and normalized ALCAM expression of endocardial cells at the AVC (Fig. EV2N–R). To visualize levels of *klf2a* expression in this experiment, we used the transgenic reporter *TgBAC(klf2a:Citrine)^mu107* (Fig. EV2N'–Q') and found that, upon knockdown of Cbx7a, *klf2a* expression remained high throughout the endocardium of both cardiac chambers in *krit1^ty219c* mutants (Fig. EV2P') when compared with wild-type (Fig. EV2N'). These experiments demonstrated that the loss of Cbx7a suppresses endothelial phenotypes in *ccm2^m201* and *krit1^ty219c* mutants and that this suppression does not involve a transcriptional normalization of Klf2a levels.

To assess whether CBX7 plays similar roles in CCM-deficient human endothelial cells, we depleted CCM2 in human umbilical vein endothelial cells (HUVECs) with an siRNA-based approach. A loss of CCM2 in these cells disrupted cell-cell junctions as exemplified by a weaker VE-cadherin labeling and holes in-between cells (Lisowska et al, 2018), and cells elongated with increased amounts of actin stress fibers (Fig. 2H,I') (Whitehead et al, 2009; Stockton et al, 2010). Strikingly, the siRNA-based knockdown of CBX7 suppressed the elongated morphologies of CCM2-depleted HUVECs, decreased the density of actin stress fibers, restored the cortical actin cytoskeleton, and increased the tightness of the endothelial cell layer to normal levels as assayed with a trans-endothelial permeability barrier assay (Fig. 2J,J',L). In comparison, an siRNA-based knockdown of CBX2, another PRC1 protein, did not alleviate CCM phenotypes in HUVECs (Fig. 2K,K',L), indicating that CBX2 is not involved in the morphology and pathophysiology in CCM2-depleted HUVECs.

## Levels of *cbx7a* mRNA expression are sensitive to blood flow and Klf2a

Biomechanical signaling via Klf2 has been implicated in both, normal physiological development of the AVC region of the heart and conditions when a loss of CCM proteins causes a defective valvulogenesis (Donat et al, 2018; Goddard et al, 2017). To determine the relationship between Cbx7a and Klf2a in the context of zebrafish *ccm2* mutants, we tested whether Klf2 contributes to the elevated expression levels of *cbx7a* mRNA. First, we generated *ccm2^m201;klf2a^sh317;klf2b^pbb42* triple mutant embryos and performed whole-mount in situ hybridizations. We found that 80% of these triple mutants had functional hearts in which the levels of *cbx7a* mRNA expression were reduced within the endocardium (Fig. 2M–O; n = 8/10 embryos with reduced expression of *cbx7a* mRNA). In 20% of *ccm2^m201;klf2a^sh317;klf2b^pbb42* triple mutant embryos, the hearts remained dysfunctional, lacked blood flow, and the expression of *cbx7a* mRNA remained elevated (Fig. 2P; n = 2/10 embryos).

Next, we assessed whether Klf2a was sufficient to induce *cbx7a* mRNA expression. We performed whole-mount in situ

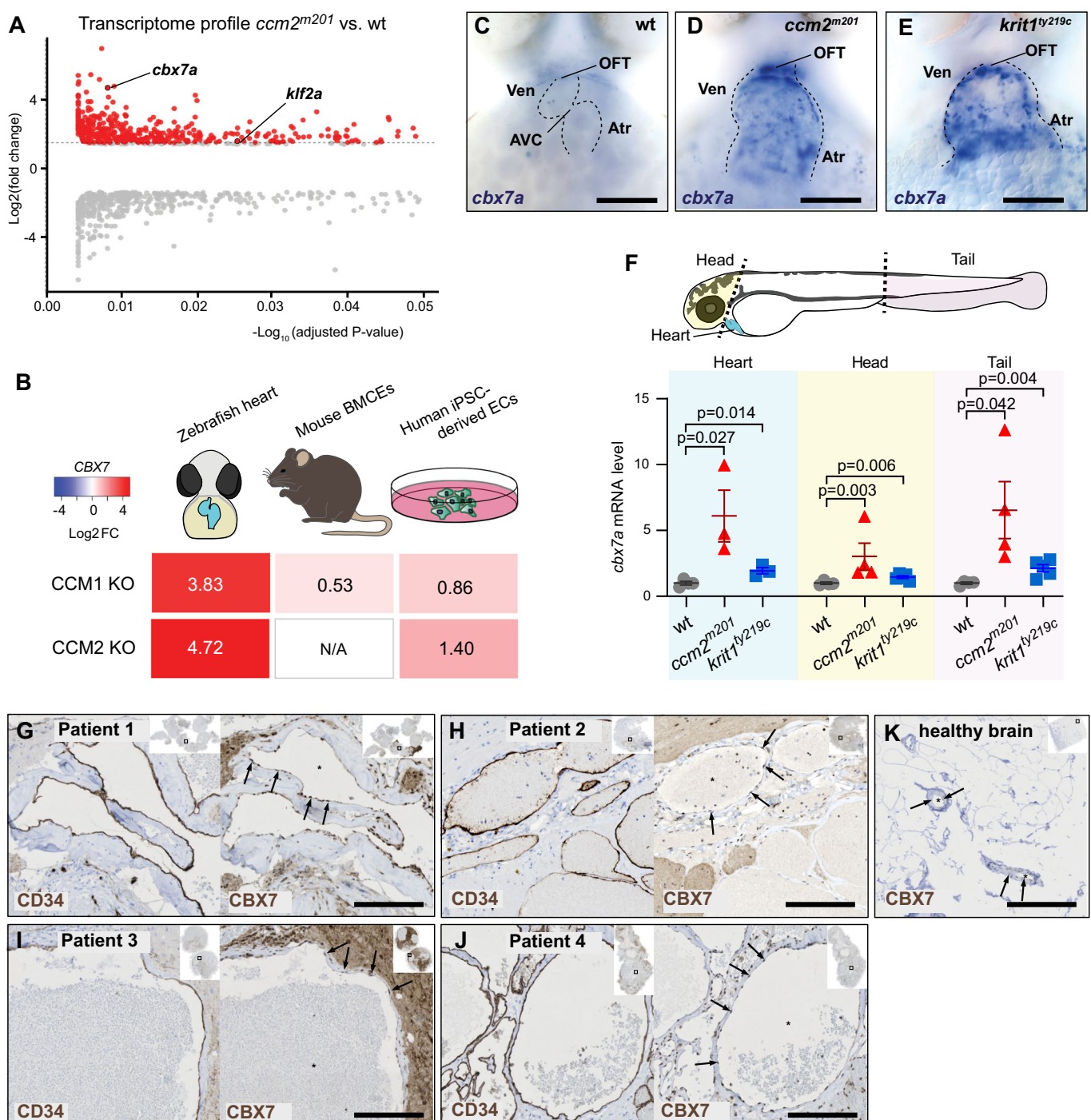

hybridization in *Tg(fli1a:Gal4FF)*^ubs3^;*Tg(UAS:klf2a)*^ig1^ double transgenic embryos that have a forced expression of Klf2a specifically within endothelial cells (Fig. 2Q, R). At 56 hpf, the overexpression of Klf2a caused a ballooning of the heart (Fig. 2Q). Some of these embryos also lacked blood flow due to a defective cardiac morphology and failure of the AVC to narrow. Notably, *Tg(fli1a:Gal4FF)*^ubs3^;*Tg(UAS:klf2a)*^ig1^ embryos without blood flow had markedly increased *cbx7a* mRNA levels (Fig. 2R). We replicated this experiment in embryos subjected to antisense

morpholino oligo injections targeting the mRNA encoding the sarcomeric protein Troponin T2a (Tnnt2a), which is essential for cardiac contractility. These *tnnt2a* morphants exhibited a complete absence of blood flow, and, in *Tg(fli1a:Gal4FF)*^ubs3^;*Tg(UAS:klf2a)*^ig1^ embryos, produced a fully penetrant cardiac ballooning phenotype (*n* = 40/40 embryos), enhanced *cbx7a* mRNA levels (Fig. 2S), and caused a loss of ALCAM expression in endocardial AVC cells (Fig. EV3A–D). These experiments demonstrated that the expression of Cbx7a is triggered by Klf2 and suppressed by blood flow.

**Figure 1.  Cbx7a/CBX7 is upregulated in different CCM disease models and lesion material of familial CCM patients.**

(A) Volcano plot of microarray data of differentially-expressed genes (calculated with the limma R package (Ritchie et al, 2015) with moderated T-statistic, adjusted *p* value <0.05 and fold change >1.4) in zebrafish *ccm2^m201* mutants (Renz et al, 2015). (B) Shown are expression fold-changes of *Cbx7a/CBX7* mRNA in different CCM disease models. Elevated mRNA expression levels of zebrafish *cbx7a* in *krit1^ty219c* and *ccm2^m201* mutants (RNA-sequencing of extracted hearts at 48 hours post fertilisation (hpf)), murine *Cbx7* in *Ccm1/Krit1^ECKO*-derived brain microvascular endothelial cells (BMECs) (Koskimäki et al, 2019), and *CBX7* in CCM1 and CCM2-deficient human iPSC-derived ECs (ECs) (RNA-sequencing). (C–E) Whole-mount in situ hybridization of zebrafish embryo at 56 hpf revealed low levels of *cbx7a* mRNA in wild-type (wt) embryos (C) and elevated levels in *ccm2^m201* (D) and *krit1^ty219c* (E) mutants throughout the entire heart. The constriction of the atrioventricular canal (AVC) was lost in *ccm2^m201* (D) and *krit1^ty219c* (E) mutants. (F) Quantifications of *cbx7a* mRNA levels by qRT-PCR from isolated tissue preparations of head, heart, or tail regions in *ccm2^m201* and *krit1^ty219c* mutants as compared to wild-type (wt) (*n* = 3–4 biological replicates per group, each replicate contained 10–15 pooled tissue samples). Statistical testing is based on the student's *T*-test (*p* values and s.e.m. bars indicated). (G, K) Immunohistochemistry stainings for CBX7 expression. CD34 marks ECs. Only low expression of CBX7 is detected in ECs of healthy brain (K, arrows). In lesions of CCM1 (G, H, J) and CCM2 (I) familial patients, CBX7 staining is markedly increased in ECs (arrows). Asterisks indicate vessel lumen. Ven ventricle, Atr atrium, OFT out flow tract. Experiments in zebrafish were done in three biological replicates. Scale bars are (C–E) 100 μm; (G–K) 200 μm. Source data are available online for this figure.

## Cbx7a is required for the activation of several Klf2 target genes in zebrafish CCM models

Next, we aimed at elucidating whether the PRC1 component Cbx7a had an effect on Klf2 target gene activation. In pursuit of this objective, we devised a transcriptomic approach to identify Cbx7a-dependent target genes of Klf2a that (1) exhibit an upregulation in *ccm* mutants, (2) show at least partial normalization in *ccm2*;*cbx7a* double mutants, and (3) are inducible by Klf2. First, we performed bulk RNA-sequencing of wild-type, *ccm2^m201* mutant, and *ccm2^m201*;*cbx7a^pbb62* double mutant hearts at 56 hpf. We expected that genes relevant to the development of *ccm* phenotypes would be normalized in both wild-type and *ccm2^m201*;*cbx7a^pbb62* double mutants. When comparing the cardiac transcriptome datasets, we found 1330 genes upregulated in *ccm2^m201* compared to wild-type. Of these, 429 genes were overlapping with genes that were also upregulated in *ccm2^m201* mutants compared to *ccm2^m201*;*cbx7a^pbb62* double mutants. Gene ontology analyses of this set of 429 genes indicated an enrichment of terms related to angiogenesis, cell migration, endothelial proliferation, endoMT, Wnt signaling, and other processes, some of which had previously been associated with the CCM pathology (Fig. 3A) (Maddaluno et al, 2013; Boulday et al, 2009, 2011; Rödel et al, 2019; Otten et al, 2018; Abdelilah-Seyfried et al, 2020; Renz et al, 2015). Conversely, 1413 genes were downregulated in *ccm2^m201* compared to wild-type, of which 513 genes were also downregulated in *ccm2^m201* compared to *ccm2^m201*;*cbx7a^pbb62* double mutants. This set of 513 genes was enriched for gene ontology terms related to cell adhesion, ECM organization, and EMT processes (Fig. 3A). Among the genes associated with the biological processes angiogenesis, canonical Wnt signaling and endoMT were several that had previously been implicated in KLF2/4 signaling (Fig. 3B) (Dekker et al, 2005; Sangwung et al, 2017). To address the relevance of these molecular pathways downstream of Klf2, we selected three of these genes as candidates for further functional studies.

## Klf2a-Wnt9b signaling mitigates CCM cardiovascular defects in zebrafish mutants

Wnt9b is directly regulated by Klf2 during mouse cardiac valvulogenesis (Goddard et al, 2017). We found that zebrafish *wnt9b* and its paralogous *wnt9a* mRNA were upregulated in *ccm2^m201* (Fig. 3B,C) and *krit1^ty219c* mutants (Fig. 3C). This *wnt9b* mRNA upregulation was also detected by whole-mount in situ hybridization in *krit1^ty219c* and *ccm2^m201* mutant hearts (Fig. EV4A,B,D,E,D',E'). To assess whether *wnt9b* was transcriptionally regulated by Klf2a, we used *Tg(fli1a:Gal4FF)^ubs3*;*Tg(UAS:klf2a)^ig1* double transgenic embryos to drive endothelial-specific overexpression of Klf2a (Renz et al, 2015). We then carried out whole-mount in situ hybridization experiments and found that zebrafish *wnt9b* mRNA was upregulated throughout the endocardium (Fig. 3E) when compared to wild-type (Fig. 3D), which confirmed that it is induced by Klf2a. Consistently, the upregulation of *wnt9b* mRNA in *ccm2^m201* and *krit1^ty219c* mutants decreased upon antisense morpholino oligo-mediated knockdown of *klf2a* and *klf2b* (Renz et al, 2015) (Fig. EV4C,F).

To test its potential role in the expression of CCM cardiovascular defects, we knocked down *wnt9b* in *ccm2^m201* and *krit1^ty219c* mutant embryos using a well-established antisense morpholino oligo against *wnt9b* (Paolini et al, 2021). The knockdown alleviated many of the cardiovascular defects associated with *ccm2^m201* and *krit1^ty219c* mutants. For instance, the loss of Ccm2 or Krit1 is strongly associated with a cardiac ballooning phenotype and loss of ALCAM at the endocardial AVC when compared with wild-type (Figs. 3F,G' and EV4G,H'). These phenotypes were clearly restored upon knockdown of *wnt9b* (Figs. 3H,H' and EV4I,I'). Similarly, in *ccm2^m201*; *wnt9b^sa20083* double mutant embryos, cardiac ballooning and ALCAM expression at the endocardial AVC were normalized (Fig. EV4K–M, K'–M'). In comparison, *wnt9b* morphants and *wnt9b^sa20083* mutants did not show any major phenotypic defects at the AVC (Figs. 3I,I' and EV4J,J',N,N'). This suggested that Klf2a-Wnt9b signaling contributes to the expression of CCM cardiovascular defects in zebrafish models of CCM.

To perform transcriptomic and epigenetic analyses, we next generated human iPSC-derived ECs that lack CCM2 (Fig. 3J–M; Appendix Fig. S2). These cells expressed the endothelial marker von Willenbrand Factor (vWF) (Appendix Fig. S2D,E) and displayed normal karyotyping profiles, indicating healthy clones (Appendix Fig. S2F,G). The CCM2-depleted iPSC-derived ECs exhibited increased amounts of actin stress fibers when compared with wild-type cells (Fig. 3J,K) and a weakened VE-cadherin labeling, indicative of weakened cell adhesion (Fig. 3L,M), which are two hallmark features of the CCM pathology. Subsequently, we used these CCM2-depleted together with wild-type control iPSC-derived ECs for RNA-seq analyses and detected increased expression levels of *CBX7, WNT9B, KLF2,* and *KLF4* mRNA (Figs. 3N and EV5A; Dataset EV2). This gene set also revealed enrichment of gene ontology terms related to angiogenesis, Rho protein signaling, cell migration, as well as a negative regulation of cell adhesion (Fig. EV5B).

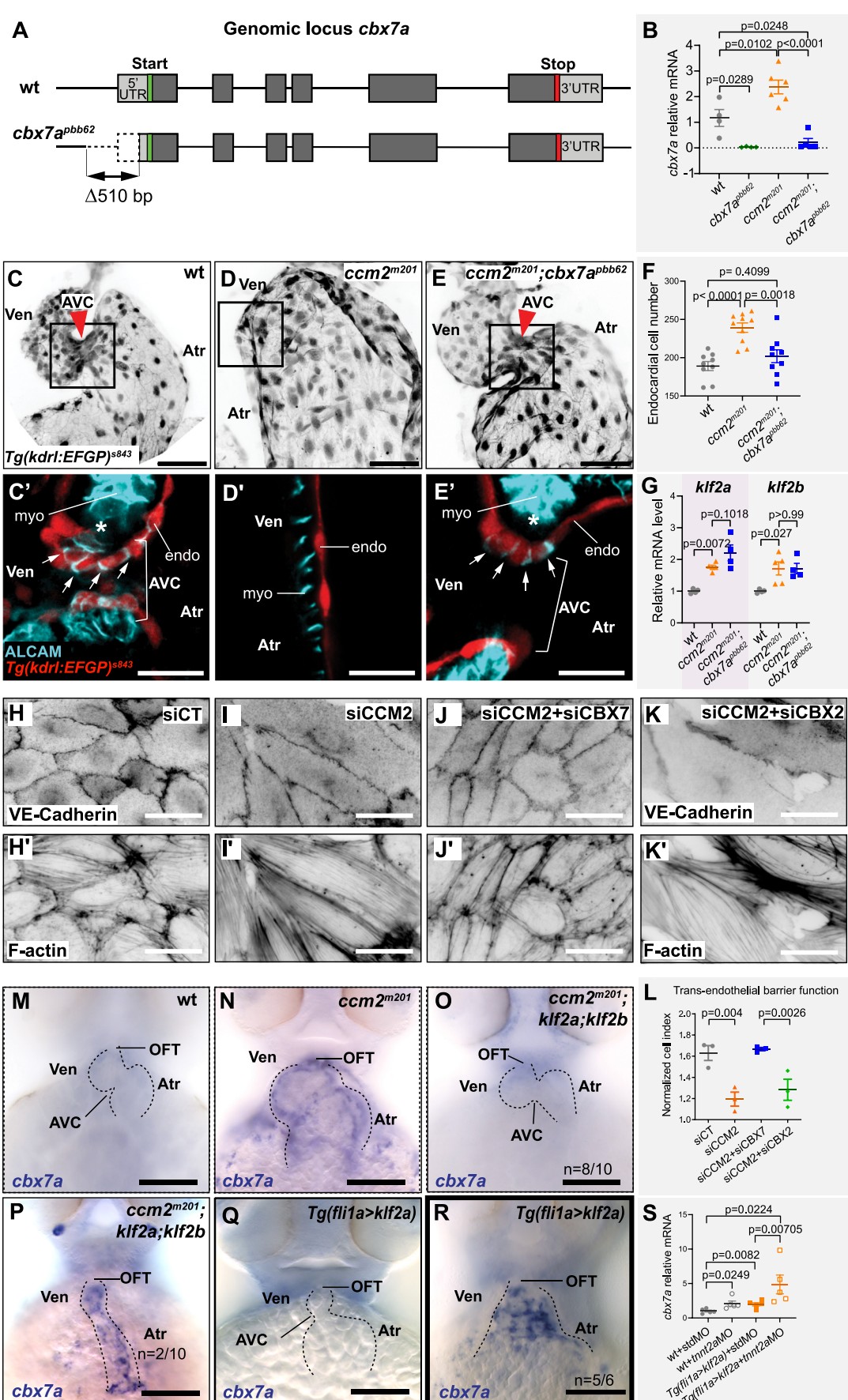

**Figure 2. Cbx7a/CBX7 suppresses CCM-associated phenotypes in zebrafish *ccm2^m201* mutants and is controlled by Klf2a and blood flow.**

(A) Schematic representation of the *cbx7a* genomic locus in zebrafish. The *cbx7a^pbb62* allele comprises a 510 bp deletion 80 bp upstream of the starting ATG (green bar). Areas in dark gray indicate the coding region and light gray areas highlight 5′ and 3′ untranslated regions (UTR). (B) Quantification of *cbx7a* mRNA levels show that its expression is completely depleted in *cbx7a^pbb62* mutants ($p = 0.0289$) and in *ccm2^m201;cbx7a^pbb62* double mutants ($p = 0.0248$). In *ccm2^m201* mutants, *cbx7a* mRNA levels are upregulated ($p = 0.0102$). Four to six replicates per group, pooled 8–10 embryos per each replicate. Statistical testing is based on one-way ANOVA with Tukey's multiple comparisons test (s.e.m. bars indicated). (C–E) Endocardium at 56 hpf, marked by *Tg(kdrl:EGFP)^s843* expression in wild-type (wt) (C), *ccm2^m201* (D), and *ccm2^m201;cbx7a^pbb62* double mutants (E). The endocardium forms the AVC (red arrowhead) between the ventricle and the atrium at 56 hpf (C). Endocardial cells of the AVC are marked by the expression of the cell junctional protein ALCAM (C′). The asterisk indicates the location of cardiac jelly between the myocardium and endocardium (C′). In *ccm2^m201* mutants, there is no AVC constriction (D) and endocardial cells lack ALCAM expression, and the cardiac jelly is markedly reduced (D′). The endocardial defects in zebrafish *ccm2^m201* mutants were suppressed when Cbx7a is genetically depleted (E). *ccm2^m201;cbx7a^pbb62* double mutants form an AVC constriction (E), and ALCAM expression (E′) and cardiac jelly (E′, asterisk) are restored. (F) Quantifications demonstrate that *ccm2^m201* mutants have higher endocardial cell numbers ($p < 0.0001$) and there is a normalization in *ccm2^m201;cbx7a^pbb62* double mutants ($p = 0.4099$ compared to wild-type; $p = 0.0018$ compared to *ccm2^m201*). Each data point represents endocardial cell numbers counted from a single heart. Statistical testing is based on the student's *T*-test (s.e.m. bars indicated). (G) Expression levels of *klf2a* and *klf2b* mRNA are elevated in *ccm2^m201* and *ccm2^m201;cbx7a^pbb62* double mutants. Shown are quantifications of whole embryos by qRT-PCR. mRNA levels of *klf2a* and *klf2b* are increased in *ccm2^m201* mutants as compared to wild-type ($p = 0.0072$ and $p = 0.027$, respectively) and do not normalize in *ccm2^m201;cbx7a^pbb62* double mutants ($p = 0.1018$ and $p = 0.99$ for *klf2a* and *klf2b*, respectively; $n = 4$–5 replicates per each group; 8–10 embryos were collected for each replicate). Statistical testing is based on student's *T*-test between mutants and corresponding wild-type siblings (s.e.m. bars indicated). (H–K) Representative images of monolayers of HUVECs silenced for control (siCT) (H, H′), CCM2 (I, I′), co-silencing of CCM2 and CBX7 (J, J′), and co-silencing of CCM2 and CBX2 (K, K′). Phenotypes of CCM2-depleted HUVECs show an elongated morphology with VE-cadherin at cell-cell junctions and increased stress fiber formation (I, I′) as compared to controls (H, H′). Restoration of cell morphology with cortical actin and reduced stress fibers is observed upon knock-down of CBX7 (J, J′), but not CBX2 (K, K′). (L) Quantification of the normalized impedance of HUVEC monolayers 24 h after serum starvation. Knockdown of CCM2 in HUVECs causes a reduction of barrier function, which is restored upon co-depletion of CBX7, but not with co-depletion of CBX2. There are three replicates per each group and statistical testing is based on one-way ANOVA with Tukey's multiple comparisons test (s.e.m bars indicated). (M–R) Representative images of *cbx7a* expression detected by whole-mount in situ hybridization in the zebrafish heart at 56 hpf. (M) *cbx7a* expression is low in wild-type and (N) expanded throughout the entire endocardium in *ccm2^m201* mutants. (O) Higher levels of *cbx7a* mRNA in *ccm2^m201* mutants are reduced in *ccm2^m201; klf2a^sh317; klf2b^pbb42* triple mutants with normal heart morphology ($n = 8/10$ triple mutants with reduced *cbx7a* mRNA expression) and elevated in triple mutants in which the cardiac morphology has not been restored (P). In *Tg(fli1a:Gal4FF)^ubs3;Tg(UAS:klf2a)^ig1* double transgenic embryos [*Tg(fli1a>klf2a)*], in which blood flow is present, mRNA levels of *cbx7a* remain low (Q). Upon endothelial-specific overexpression of Klf2a and concomitant loss of blood flow, *cbx7a* mRNA levels are elevated (R). (S) Quantification of *cbx7a* mRNA levels by qRT-PCR in *Tg(fli1a>klf2a)* embryos that also lack blood flow (*tnnt2a* MO against sarcomeric protein Tnnt2a) or with normal cardiac function (standard morpholino, std MO). There are five replicates per each group and each replicate is based on 8–10 pooled embryos. Statistical testing is based on student's *T*-test (s.e.m. bars indicated). AVC atrioventricular canal, endo endocardium, myo myocardium, OFT outflow tract, Ven ventricle, Atr atrium. Red arrowheads indicate the location of the AVC and white arrows indicate the presence of ALCAM expression in AVC endocardial cells. Experiments in zebrafish were done in three biological replicates. Scale bars are (M–R) 100 µm, (C–E, H–K, H′–K′) 50 µm, (C′–E′) 20 µm. Source data are available online for this figure.

Proteins of the PRC2 are involved in the tri-methylation of histone H3 at Lys27 (H3K27me3). This recruits PRC1 proteins to defined chromatin domains, which results in H2AK119ub modifications and causes a transcriptional downregulation of genes (Blackledge and Klose, 2021). CUT&RUN-seq analysis of H3K27me3 marks in CCM2-deficient iPSC-derived ECs, as compared to their wild-type control ECs, revealed a notable reduction of this histone modification. Consequently, a predicted transcriptional activation was observed at loci associated with blood vessel morphogenesis, vasculogenesis, VEGF receptor signaling (all related to angiogenesis), and the Rho protein signaling pathway (related to stress fiber formation) (Fig. EV5C). We compared these GO terms to H3K4me3 marks, which are mainly enriched in actively expressed genes, and noticed a robust convergence with terms associated with EC migration, angiogenesis, and Rho protein signaling (Fig. EV5D). These findings suggest that the loss of CCM2 affects the activity of PRC1/2 and is associated with CCM-related pathological phenotypes, affecting angiogenesis, Rho protein signaling, and EC migration, among others.

Consequently, we tested whether this gene might be relevant to human pathology by investigating whether its transcripts were elevated in primary human CCM lesion material. Indeed, mRNA expression levels of *WNT9B* and its orthologue *WNT9A* mRNA were increased in lesion material from both sporadic and familial cases of CCM (Fig. 3N). Taken together these data show that Cbx7a-dependent upregulation of Wnt9b contributes to CCM-related phenotypes in zebrafish models of CCM. Furthermore, transcriptomic profiles of human iPSC-derived EC and lesion material from CCM patients revealed an upregulation of *WNT9B*, suggesting it may be involved in the CCM pathology.

## Tie1/2 signaling contributes to the expression of cardiovascular defects in zebrafish *ccm2* mutants

Whole-transcriptome analyses demonstrated that Cbx7a promotes an increased expression of several genes involved in angiogenesis signaling (Fig. 3A,B). For instance, *ccm2^m201* mutants exhibited an upregulation of the genes encoding the Angiopoietin receptor Tie2 and its activating ligand Angiopoietin-1 (Angpt1). Their gene expression was normalized in *ccm2^m201;cbx7a^pbb62* double mutants (Fig. 3B,O). The loss of only Cbx7a slightly reduced expression levels of *angpt1* and *tek* mRNA encoding Tie2 (Fig. 3P,Q). We also found that the *angpt1* and *tek* transcript levels were upregulated in *Tg(fli1a:Gal4FF)^ubs3;Tg(UAS:klf2a)^ig1* double transgenic embryos, which have an increased endothelial expression of Klf2a (Fig. 3P,Q). The upregulation of *tek* was normalized to wild-type levels upon knockdown of Cbx7a (Fig. 3Q), while levels of *angpt1* were not significantly reduced (Fig. 3P).

To address the functional relevance of increased Tie2 signaling for the *ccm2* cardiovascular phenotype, we lowered its activity in CCM models of zebrafish by using a pharmacological intervention strategy and through genetic depletion. We treated embryos between the 16-somite stage and 48 hpf with 10 µM of BAY826, an antagonistic inhibitor of the two Angiopoietin receptors Tie1 and Tie2 (Schneider et al, 2017), or with DMSO as a control (Fig. 3R–U). This inhibitor treatment suppressed *ccm2* mutant cardiac ballooning and the loss of ALCAM expression at the endocardial AVC (Fig. 3S-T'), while there was no effect on wild-type embryos (Fig. 3U,U'). Next, we used a genetic depletion approach and injected a control antisense morpholino oligo or one against *krit1* into in-crosses from *tek^hu1667/+;tie1^bns208/+* heterozygous

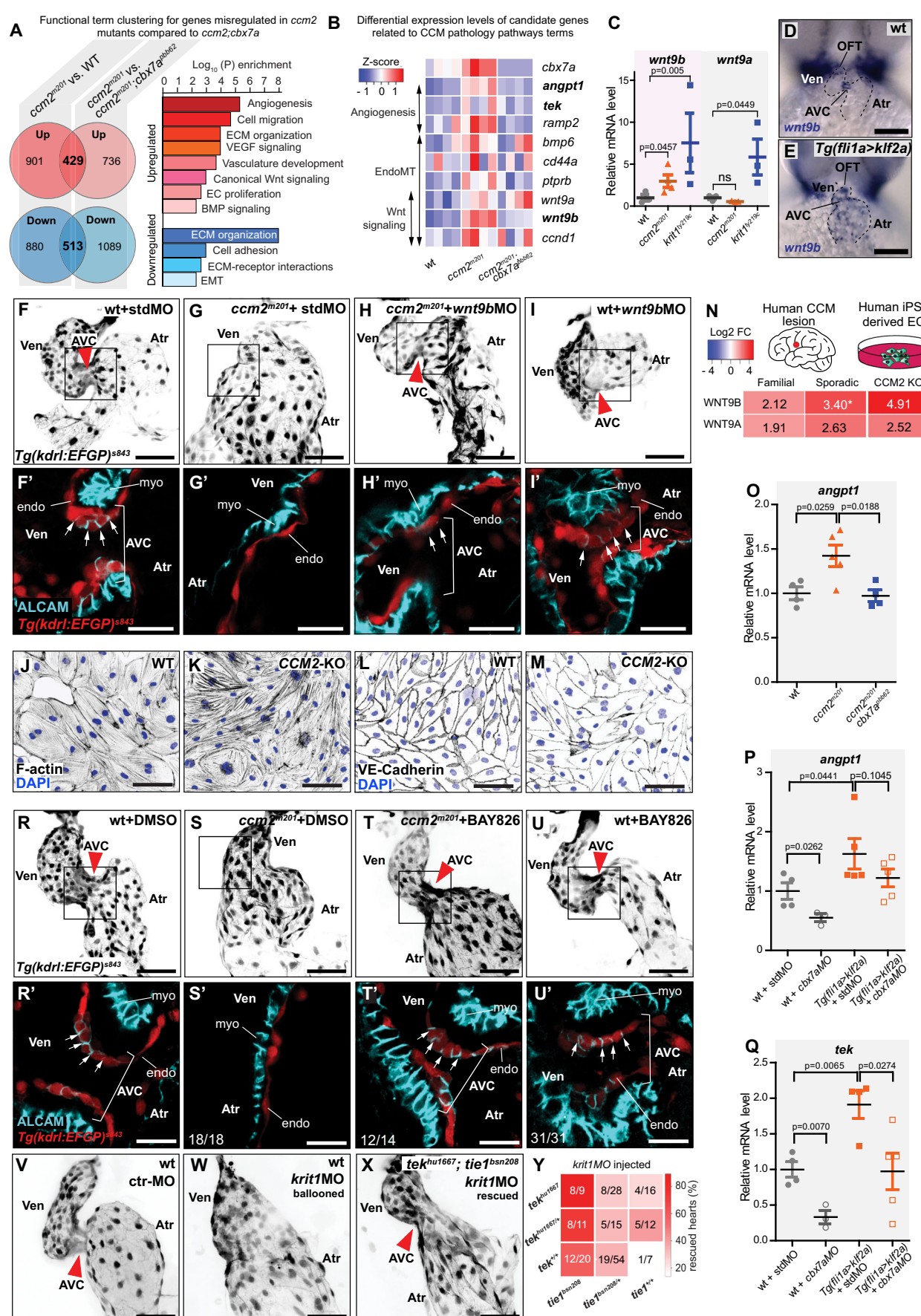

**Figure 3. Cbx7a/CBX7 contributes to pathological gene expression in CCM, impacting cardiovascular defects in zebrafish mutants.**

(A) Functional clustering of gene ontology terms for deregulated genes identified in RNA-sequencing datasets from *ccm2^m201^* in comparison with wild-type (wt) and *ccm2^m201^;cbx7a^pbb62^* double mutants in comparison with *ccm2^m201^* mutants. Differentially-expressed genes were identified by using DEseq2 (Galaxy Version 2.11.40.7) and a Wald statistic for pairwise comparisons. (B) Schematic overview of candidate genes related to CCM pathology gene ontology terms. Shown are fold-changes of expression levels in wild-type, *ccm2^m201^*, and *ccm2^m201^;cbx7a^pbb62^* double mutants. (C) Quantification of *wnt9b* and *wnt9a* mRNA levels in *ccm2^m201^* and *krit1^ty219c^* extracted hearts. *wnt9b* is upregulated in both mutants while *wnt9a* is upregulated only in *krit1^ty219c^* mutants ($n = 3$–4 replicates per each group, and each replicate is a pool of 50–100 hearts). Statistical testing is based on one-way ANOVA with Tukey's multiple comparisons testing (s.e.m. bars indicated). (D, E) Representative images of whole-mount in situ hybridizations of *wnt9b* expression in the 56 hpf zebrafish heart. In wild-type, *wnt9b* is expressed in a defined domain confined to the atrioventricular canal (AVC) (D). In *Tg(fli1a:Gal4FF)^ubs3^;Tg(UAS:klf2a)^ig1^* double transgenic embryos [*Tg(fli1a>klf2a)*] with an endothelial-specific overexpression of Klf2a, the expression of *wnt9b* is expanded into the cardiac chambers (E). (F–I) Functional rescue of endocardial phenotypes through genetic depletion of Wnt9b in zebrafish *ccm2^m201^* mutant embryos. Shown are maximum projections of confocal image z-stacks of zebrafish hearts at 56 hpf, with the endocardium being marked by *Tg(kdrl:EGFP)^s843^* expression. (F) The AVC region forms correctly in wild-type hearts (red arrowhead). The expression of ALCAM in endocardial AVC cells (F'; white arrows) is lost in *ccm2^m201^*mutants (G'). Upon depletion of *wnt9b* using an antisense oligo morpholino, the overall morphology of *ccm2^m201^* mutant hearts is improved, the AVC region restored (H; red arrowhead), and ALCAM expression at the endocardial AVC is restored (H'; white arrows). Knock-down of *wnt9b* alone does not affect AVC formation (I; red arrowhead) and ALCAM expression (I'; white arrows). (J–M) Representative fluorescence microscopy images of CCM2-deficient (CCM2-KO) and wild-type (WT) iPSC-derived ECs immunostained for F-Actin (J, K), and VE-Cadherin (L, M). (N) Schematic overview of WNT9A and WNT9B mRNA levels in primary human lesion material from familial and sporadic CCM patients as compared to healthy brain material and in human iPCS-derived endothelial cells depleted of CCM2 as compared to non-edited cells. In all cases, *WNT9A* and *WNT9B* are upregulated. (O–Q) Quantifications based on qRT-PCR of *angpt1* and *tek/tie2* mRNA levels in *ccm2^m201^* and *ccm2^m201^;cbx7a^pbb62^* double mutants (O) and in *cbx7a* antisense morpholino-injected *Tg(fli1a>klf2a)* (P, Q) ($n = 4$–5 replicates and each replicate contains 8–10 pooled embryos). Statistical testing is based on student *T*-test (s.e.m. bars indicated). (R, X) Shown are maximum projections of confocal image z-stacks of zebrafish hearts at 48 hpf in which the endocardium is marked by *Tg(kdrl:EGFP)^s843^* expression. Wild-type embryos treated with DMSO form a normal AVC (R; red arrowhead) and express ALCAM in endocardial cells at the AVC (R'; white arrows). Endocardial defects in *ccm2^m201^* mutants include the loss of the AVC constriction (S) and ALCAM expression (S'). (T) *ccm2^m201^* mutant embryos treated with the Tie1/2 signaling inhibitor BAY826 have a normalized cardiac morphology, show a restoration of the AVC region (T; red arrowhead) and endocardial ALCAM expression (T'; white arrows). (U) Treatment of wild-type embryos with BAY826 has only a mild effect on cardiac morphology as AVC (U; red arrowhead) and expression of ALCAM in endocardial AVC cells (U'; white arrows) are present. (V–X) The genetic depletion of zebrafish *tek* and *tie1* genes suppresses the cardiac ballooning phenotype in *krit1* morphants. At 56 hpf, wild-type embryos injected with an antisense morpholino against *krit1* lose that AVC constriction and exhibit strongly ballooned hearts, which is a characteristic feature of the cardiac phenotype in zebrafish *ccm* mutants (W). The genetic depletion of *tek/tie2* and *tie1* in *krit1* morphants restores the AVC constriction and suppresses cardiac ballooning (X, red arrowhead). (Y) Quantifications of the *krit1* morphant cardiac phenotype in different *tek^hu1667^;tie1^bns208^* mutant combinations. The penetrance of *krit1* morphant cardiac phenotypes decreases with lowered copy numbers of wild-type *tek/tie2* and *tie1* alleles and is most strongly suppressed by the combined loss of both Tie1/2 receptors. Numbers in boxes are embryos with a rescued heart / total number of embryos with the respective genotype. AVC atrioventricular canal, endo endocardium, myo myocardium, Ven ventricle, Atr atrium. Red arrowheads indicate AVC, while white arrows indicate the presence of ALCAM expression in AVC endocardial cells. Experiments in zebrafish were done in three biological replicates. Scale bars are (D, E, J–M) 100 μm, (F–I; R–U; V–X) 50 μm, (F'–I'; R'–U') 20 μm. Source data are available online for this figure.

parents (Fig. 3V–Y). This in-cross produced a subset of *tek^hu1667^;tie1^bns208^* homozygous mutants and allowed us to assess the relevance of Tie1/Tie2 signaling on the expression of the *krit1* loss-of-function phenotype. The cardiac ballooning phenotype in *krit1* morphants decreased significantly in the presence of mutant alleles, particularly of *tie1*, and *krit1* phenotypes were most strongly suppressed in *tek^hu1667^;tie1^bns208^* double mutants (Fig. 3W–Y). This demonstrated that Tie1/Tie2 signaling is involved in cardiovascular phenotypes in zebrafish models of CCM.

## CBX7 presents a promising novel target for pharmacological intervention in the treatment of CCM

There is an urgent need for an effective pharmacological strategy against CCM, which led us to investigate the potential of targeting Cbx7. Our observation that the expression of *cbx7a* mRNA significantly increases in zebrafish models of CCM, suggested that this protein holds some promise as a target for pharmacological intervention. Two well-characterized small compound CBX7 inhibitors, MS351 and MS37452, have previously been tested in the treatment of prostate cancer (Ren et al, 2015, 2016) (Fig. 4A). To test whether Cbx7a is an effective pharmacological target in CCM, we treated *ccm2^m201^* mutant zebrafish embryos between 16-54 hpf with either MS351 or MS37452 and assessed whether this had a preventive effect on the formation of cardiac morphological defects that are characteristic of the embryonic loss of function phenotypes in zebrafish (Renz et al, 2015; Mably et al, 2006; Donat et al, 2018). Indeed, treatment of zebrafish *ccm2^m201^* mutants under this regiment suppressed the exaggerated ballooning phenotype of the heart

(Fig. 4B–D, $n = 9/20$ mutant embryos rescued with MS351; $n = 4/22$ mutant embryos rescued with MS37452) and restored atrioventricular differentiation in a subset of treated embryos as assessed by the expression of ALCAM within endocardial cells (Fig. 4B'–D', arrows).

Next, we tested this pharmacological intervention strategy in a pre-clinical mouse model with an endothelial-specific inducible knockout construct for *CCM3* (Bravi et al, 2015; Wang et al, 2010). We chose CCM3 for this pre-clinical study because this presents as the most aggressive form of CCM in patients. Upon Tamoxifen-induction at postnatal day 1 (P1), Cre-positive endothelial-specific *CCM3* knock-out mice (*CCM3^iECKO^*) received a daily intragastric injection of MS37452 (2 mg/g body weight). Subsequently, the lesion burden was assessed upon tissue collection at postnatal day 8 (P8), with respect to the total lesion number and area (Fig. 4E–J). We found that MS37452-treated *CCM3^iECKO^* mice showed lower lesion numbers and area (Fig. 4G–J) compared with untreated animals. This intervention of a pharmacological blockade against an epigenetic regulator in CCM demonstrates that CBX7 is a promising new therapeutic target in CCM.

## Discussion

Our work and a second recent publication (Valentino et al., 2024) demonstrates that epigenetic regulation by Polycomb repressive complexes affects pathophysiological transcription in the vascular malformation disease CCM (Valentino et al, 2024). Here, we demonstrate that CBX7, a component of PRC1, is upregulated under pathophysiological conditions associated with CCM. This

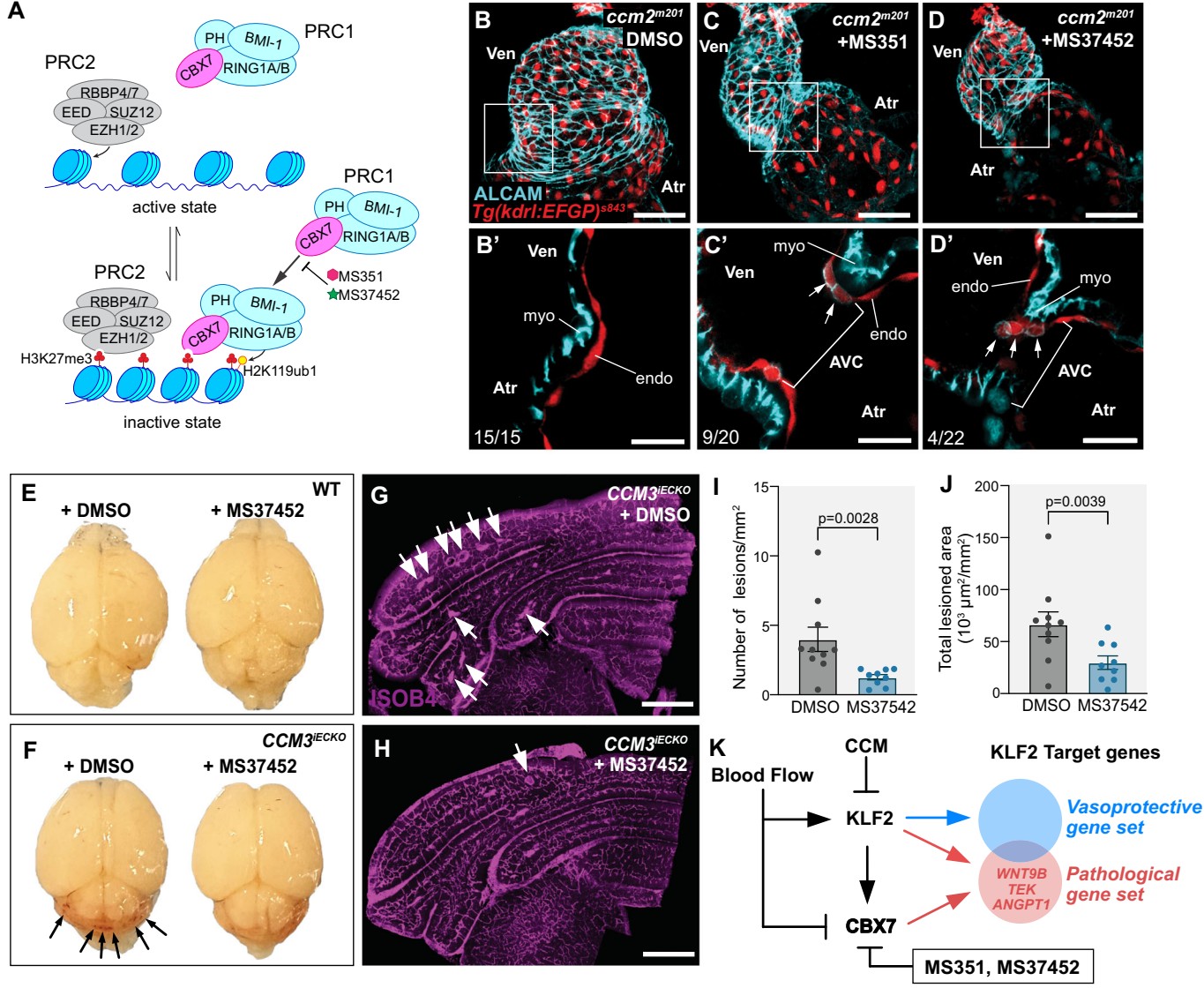

**Figure 4. The pharmacological inhibition of PRC1 protein Cbx7a/CBX7 suppresses CCM-associated cardiac phenotypes in zebrafish *ccm2^{m201}* mutants and lesion burden in a murine pre-clinical CCM3 disease model.**

(A) Proteins of the PRC2 are involved in the tri-methylation of histone H3 at Lys27 (H3K27me3). This epigenetic modification recruits CBX7 and other PRC1 proteins to these genomic sites. The RING1A/B ubiquitin ligases of the PRC1 catalyze H2AK119ub modifications, which results in transcriptional downregulation of genes. The two small compound drugs MS351 and MS37452 inhibit the activity of CBX7. (B–D) The preventive pharmacological treatment of zebrafish *ccm2^{m201}* mutants with CBX7 inhibitors suppresses the development of a cardiac ballooning phenotype and the loss of AVC constriction. Shown are maximum projections of confocal image z-stacks of zebrafish hearts at 56 hpf. The endocardium is marked by the expression of *Tg(kdrl:EGFP)^{s843}*. The *ccm2^{m201}* mutant heart is ballooned (B) and lacks expression of ALCAM in atrioventricular canal (AVC) endocardial cells. Treatment with the CBX7 inhibitor MS351 (C, C') or MS37452 (D,D') strongly improves cardiac morphology (C, D) and normalizes the endocardial expression of ALCAM in AVC endocardial cells (C',D'; arrows). (E–J) The inhibition of CBX7 with MS37452 in *CCM3^{iECKO}* (n = 4) and control mice (n = 10). (E) Treatment with MS37452 does not produce any adverse effects on the brains of the treated mice. (F) *CCM3^{iECKO}* animals treated with DMSO show hemorrhages in the region of the cerebellum (arrows). These are markedly reduced in mice treated with MS37452 (H). (G, H) Isolectin B4 (ISOB4) immunostainings reveal fewer lesions in *CCM3^{iECKO}* mice that were treated with MS37452, reducing markedly the number of lesions (G, H, arrows). (I) Quantifications of lesion numbers and (J) total lesioned area. Statistical testing is based on unpaired student's *T*-test (s.e.m bars indicated). (K) Model figure depicting the control of vasoprotective and pathophysiological gene expression through the interplay of the proteins of the CCM complex, the PRC1 complex protein CBX7, the transcription factor KLF2, and blood flow as a biophysical modulator. Under physiological conditions, CBX7 expression is negatively controlled by blood flow and the CCM complex downregulates KLF2 expression. The loss of CCM becomes detrimental in low-shear stress regions due to the activation of KLF2 expression. In addition, CBX7 expression becomes upregulated due to the lack of blood flow. Together this renders pathophysiological targets, such as *WNT9B*, *TEK*, and *ANGPT1*, available for KLF2-dependent activation. AVC atrioventricular canal, endo endocardium, myo myocardium, Ven ventricle, Atr atrium. Experiments in zebrafish were done in three biological replicates. Scale bars are (B–D) 50 μm; (B'–D') 20 μm; (G, H) 500 μM. Source data are available online for this figure.

presents CBX7 as a particularly attractive target for pharmacological intervention because its expression correlates with active phases of the CCM pathology, and inhibitory drugs are readily available (Ren et al, 2015, 2016). We show that silencing of Cbx7a/CBX7 suppresses CCM phenotypes in zebrafish cardiovascular developmental and human endothelial cells. The pharmacological suppression of Cbx7 suppresses lesion size and number in a preclinical CCM3 disease mouse model and in the acute zebrafish cardiovascular developmental CCM model. We also show that Cbx7a, which is regulated by Klf2a and blood flow, affects the pathological pattern of endothelial cell gene expression induced by Klf2a during the development of CCM-associated phenotypes.

Strikingly, Cbx7a itself is not essential during development in zebrafish since homozygous mutants survive to adulthood and are fertile. This suggests that Cbx7a activity has some particular role specifically under pathophysiological conditions. When endothelial cells experience loss-of-function mutations in the genes that encode CCM proteins, KLF2 becomes upregulated also in regions of low fluid shear stress (Renz et al, 2015; Zhou et al, 2016). Functional studies in zebrafish and mouse models showed that this triggers the expression of a pattern of genes that is pathophysiological and drives the formation of CCMs (Abdelilah-Seyfried et al, 2020). In the healthy endothelium, KLF2 is induced by high levels of fluid shear stress and triggers a pattern of gene expression that is vasoprotective (Dekker et al, 2006; Poelmann and Gittenberger-de Groot, 2018; Novodvorsky and Chico, 2014). What initiates this shift from vasoprotective to pathological gene expression, and how this is impacted by biomechanical signaling, has long been an enigma; equally unclear has been the identity of the genes downstream of KLF2 that cause the development of this pathology.

Here, we discovered that Klf2a activates the expression of the PRC1 protein Cbx7a in zebrafish models of CCM. Interestingly, this is apparently a physiological regulation since *Cbx7* is down-regulated in cardiac microvascular endothelial cells obtained from *Klf2;Klf4* double-knockout mice (Sangwung et al, 2017). Our findings show that the activation of Cbx7a facilitates the expression of target genes in a pathophysiological pattern that involves *wnt9b*, *angiopoetin1-tek*, and genes characteristic of endoMT (Fig. 4K). Furthermore, biomechanical signaling has opposing effects on expression levels of Cbx7a and Klf2a, whereby Cbx7a expression is suppressed by blood flow (Fig. 4K). Consequently, pathological gene expression in CCM results from the integrated activities of Cbx7a and Klf2a, with blood flow impacting their activities (Fig. 4K). This points to a phenomenon we demonstrated in earlier work when we found that blood flow has protective effects and prevents cardiovascular anomalies in *krit1/ccm1* mutant zebrafish (Rödel et al, 2019). Similarly, CCM lesions in patients are slow-flow anomalies.

We propose that CBX7 activation in CCM is a conserved mechanism since CBX7 protein is enriched in EC nuclei from CCM patient-derived lesion material. We also find that *Cbx7* is upregulated in a murine endothelial-specific *Krit1/Ccm1* knockout and *CBX7* is upregulated in human endothelial cell models of CCM, where it plays a crucial role in the pathological phenotype. Moreover, human homologs of zebrafish Klf2 target genes including *WNT9A/B* are also upregulated in patient-derived primary CCM lesion material. The activation of Wnt9b by Klf2 was first discovered in studies of endocardial valvulogenesis in mice

and zebrafish. This process involves an increase in canonical WNT pathway signaling, which in turn promotes an endoMT among endocardial cells (Goddard et al, 2017; Paolini et al, 2021). Our new finding lends further evidence to the role of WNT9 signaling in CCM. Previous studies found that a loss of *Krit1/Ccm1* in murine endothelial cells caused an increase in the nuclear localization of the transcriptional regulator β-catenin and signaling via the WNT pathway (Glading and Ginsberg, 2010). Similarly, an endothelial-specific knockout of *Krit1/Ccm1* in mice caused an endoMT (Maddaluno et al, 2013). The same group later showed that the small WNT inhibitory drugs sulindac sulfide and sulindac sulfone effectively reduced the burden of vascular lesions in endothelial CCM3-deficient mice (Bravi et al, 2016). These studies were called into question by Zhou and colleagues, who did not find any evidence of endoMT or increased Wnt signaling in CCM (Zhou et al, 2016). The present study strongly supports the original findings by Maddaluno and colleagues (Maddaluno et al, 2013) and provides further evidence for an increase in endoMT and Wnt signaling in CCM.

Our findings are also supported by a well-established role for PRC1 in controlling developmental EMT processes, where Wnt signaling also plays an important role (Lee et al, 2006; Teeuwssen and Fodde, 2019). For instance, CBX7 is expressed in human hematopoietic stem cells, where its targets comprise Wnt pathway genes, including the WNT9A and WNT9B receptor FZD9B (Klauke et al, 2013). To our knowledge, our findings demonstrate for the first time that the epigenetic regulation of these pathways contributes to CCM vascular malformations. It will be of great interest to clarify whether PRC1/2 components also contribute to other malformations of the vasculature.

Here, we found that treating zebrafish mutants with the competitive TIE1/2 antagonist BAY826 suppressed CCM-associated cardiovascular defects. This finding strongly suggests a molecular link between Krit1/Ccm2 and the vascular malformations caused by gain-of-function mutations in the TIE2 receptor. Previous studies in mice provided evidence that Tie2 signaling increases in endothelial-specific knockouts of *Ccm3*. In contrast, murine *Krit1* or *Ccm2* knockouts showed no changes in Tie2 activity (Zhou et al, 2021). This group also discovered that a loss of Ccm3 increased the secretion of the Tie2 low-affinity ligand Angiopoietin 2 (Zhou et al, 2016). It, in turn, acts as an antagonist of the blood vessel-stabilizing Angiopoietin-1 and can disrupt endothelial cell junctional integrity and angiogenesis (Eklund and Olsen, 2006; Maisonpierre et al, 1997). Our findings now imply that an increase in TIE2 signaling is common in endothelial cells upon a loss of any of the CCM proteins. This suggests a connection between CCMs with another group of low blood flow venous malformations caused by gain-of-function variants of TIE2, such as the common TIE2^{L914F} receptor variant (Limaye et al, 2009). In venous malformations, TIE2 gain of function variants trigger a strong activation of PI3Ka/AKT signaling, which has also been implicated in CCM (Ren et al, 2021; Peyre et al, 2021).

The development of CCMs requires that endothelial cells undergo excessive growth, which occurs when they acquire oncogenic gain-of-function mutations in *PIK3CA* (Ren et al, 2021; Peyre et al, 2021). Particularly aggressive forms of CCM have also been linked to oncogenic *MAP3K3* gain-of-function mutations that are sufficient to produce the malformations but may

also occur in conjunction together with *PIK3CA* gain-of-function or CCM loss-of-function mutations (Weng et al, 2021). The involvement of the polycomb repressive complexes 1/2 in various cancers has been well documented. Diverse components of these epigenetic regulatory complexes are involved in different cancers. CBX7, for example, is upregulated in acute myeloid leukemia and other malignancies (Parreno et al, 2022). In the former case, CBX7 binds to several H3K9 methyltransferases which possess a tri-methylated lysine peptide motif similar to its normal histone target H3K27me3 (Jung et al, 2019). Hence, the discovery that the PRC1 protein CBX7 is activated in CCM provides yet another hint at a cancer-like molecular mechanism involved in vascular malformations. Our findings suggest that activation of CBX7 causes not only an increase in TIE2-PI3Ka signaling but also pro-oncogenic properties such as endoMT. Our data from CCM2 knockout iPCS-derived endothelial cells reveal a loss of tri-methylation at sites linked to endothelial dysfunction in CCM. This suggests a shift in PRC protein activity and a potential loss of CBX7 at these sites. These issues will be of interest in further attempts to understand the underlying etiology of vascular malformations and their connection to tissue-level properties, such as the mechanical stresses induced by blood flow.

# Methods

### Reagents and tools table

| Reagent/resource | Reference or source | Identifier or catalog number |
|---|---|---|
| **Experimental models** | | |
| Zebrafish | | |
| *krit1^ty219c* | Mably et al, 2006 | |
| *ccm2^m201* | Stainier et al, 1996 | |
| *cbx7a^pbb62* | This study | |
| *klf2a^sh317* | Novodvorsky et al, 2015 | |
| *klf2b^pbb42* | Fontana et al, 2020 | |
| *tek^hu1667* | Gjini et al, 2011 | |
| *tie1^bns208* | Carlantoni et al, 2021 | |
| *Tg(fli1a:Gal4FF)^ubs3* | Herwig et al, 2011 | |
| *Tg(UAS:klf2a)^ig1* | Renz et al, 2015 | |
| *Tg(fli1:NLS-mCherry)^ubs10* | Heckel et al, 2015 | |
| *Tg(klf2a:Citrine)^mu107* | Sugden et al, 2017 | |
| *Tg(kdrl:EGFP)^s843* | Jin et al, 2005 | |
| In vitro | | |
| HUVEC | Lonza | |
| ODInCas9-hiPSCs | Lundin et al, 2020 | |
| Mouse | | |
| Cdh5(PAC)-Cre-ER^T2/ CCM3^flox/flox | Bravi et al, 2015 | |
| **Recombinant DNA** | | |
| pT3TS-nCas9n | Addgene | #46757 |

| Reagent/resource | Reference or source | Identifier or catalog number |
|---|---|---|
| **Antibodies** | | |
| Zn-8/Alcam | Developmental Studies Hybridoma Bank | AB_531904 |
| Mf20 | Developmental Studies Hybridoma Bank | AB_2147781 |
| VE-Cadherin antibody BV9 | Sigma-Aldrich | MABT129 |
| TRITC phalloidin | Sigma-Aldrich | P-1951 |
| CD31 MicroBeads | Miltenyi Biotec | 130-097-418 |
| Ve-Cadherin | Cell Signaling Technologies | D87F2 |
| VWF | Sigma-Aldrich | HPA001815 |
| Phalloidin 647 Plus | Thermo Fisher | A30107 |
| biotinylated Isolectin B4 | Vector Laboratories | B-1205 |
| CBX7 | Sigma-Aldrich | #HPA056480 |
| CD34 | Dako | #IR632 |
| phalloidin conjugates with Atto 647 | Sigma-Aldrich | 65906 |
| Streptavidin conjugated with Alexa Fluor 647 | Invitrogen | S21374 |
| Alexa Fluor-conjugated AF 488/ 546/ 633 | Invitrogen | A21206/A32723/ A-11030/ A-21052 |
| **Oligonucleotides and other sequence-based reagents** | | |
| PCR primers | This study | Table EV3 |
| Morpholinos | | |
| siRNA_CT | Dharmacon | D-001810-01-50 |
| siRNA_CCM2 | Dharmacon | L-014728-01 |
| siRNA_CBX7 | Dharmacon | L-009561-01 |
| siRNA_Cbx2 | Dharmacon | L-008357-01 |
| **Chemicals, Enzymes and other reagents** | | |
| MS37452 | Sigma-Aldrich | SML1405 |
| MS351 | Cayman Chemical | Cay19549 |
| Bay826 | Tocis | 6579 |
| Lipofectamine RNAimax | Life Technologies | 13778-150 |
| Lipofectamine CRISPRMAX Cas9 Transfection Reagent | Invitrogen | CMAX00003 |
| iPSC Single-Cell Cloning DEF-CS™ Culture Media Kit | Cellartis | Y30021 |
| HiScribe™ T7 Quick High Yield RNA Synthesis Kit | New England Biolabs | E2050S |
| mMESSAGE mMACHINE™ T3/ T7 Transcription Kit | Invitrogen | AM1348/ AM1344 |
| DIG RNA Labeling Kit | Roche | 11175025910 |
| 1-phenyl-2-thiourea (PTU) | Sigma-Aldrich | 103-85-5 |
| Quick RNA micro-prep | Zymo Research | R1050 |
| KAPA SYBR® Fast qPCR Kit | Roche | SFUKB |
| TrypLE Select | Thermo Fisher Scientific | 12563-029 |

| Reagent/resource | Reference or source | Identifier or catalog number |
| --- | --- | --- |
| Agencourt RNAdvance Cell kit | Beckman Coulter | A47943 |
| KAPA mRNA HyperPrep Kit | Roche | KK8581 |
| NEBNext® Ultra II Directional RNA Library Prep Kit for Illumina® | New England Biolabs | E7760L |
| QuickExtract DNA extraction solution | Lucigen | 101098 |
| **Software** | | |
| Imaris 9 | Oxford Instruments | |
| Prism 5/10 | GraphPad | |
| Fiji | NIH | |
| ZEN | Carl Zeiss | |
| CCTop | Stemmer et al, 2015 | |
| **Other** | | |
| qTower 3 | Analytik Jena | |
| xCELLigence real-time cell analyzer (RTCA) system | Ozyme | |
| NovaSeq6000 | Illumina | |
| Fragment Analyzer 5300 system | Agilent | |
| TapeStation system 2200 | Agilent | |
| NextSeq 550 sequencer | Illumina | |

## Zebrafish

Handling of zebrafish was done according to FELASA guidelines (Aleström et al, 2020), in compliance with German and Brandenburg state law, carefully monitored by the local authority for animal protection (LAVG, Brandenburg, Germany, Animal protocol #2347-43-2021). Zebrafish strains used in this study include *krit1*[ty219c] and *ccm2*[m201] (Stainier et al, 1996; Mably et al, 2006), *cbx7a*[pbb62] (this study), *klf2a*[sh317] (Novodvorsky et al, 2015), *klf2b*[pbb42] (Fontana et al, 2020), *tek*[hu1667] (Gjini et al, 2011), *tie1*[bns208] (Carlantoni et al, 2021), *Tg(fli1a:Gal4FF)*[ubs3] (Herwig et al, 2011), *Tg(UAS:klf2a)*[ig1] (Renz et al, 2015), *Tg(fli1:NLS-mCherry)*[ubs10] (Heckel et al, 2015), *Tg(klf2a:Citrine)*[mu107] (Sugden et al, 2017), and *Tg(kdrl:EGFP)*[s843] (Jin et al, 2005).

### Generation of transgenic and mutant fish

*cbx7a* mutants were generated by CRISPR/Cas9-mediated mutagenesis. Single-guide RNAs (sgRNAs) were designed with the software tool CCTop (https://cctop.cos.uni-heidelberg.de:8043/index.html)(Stemmer et al, 2015) including sgRNA1 and sgRNA2 (sequences in Table EV1). sgRNAs were used to excise a 510 bp fragment, 80 nt upstream of the coding sequence, to generate a promoter-less allele (*pbb62*). sgRNAs were synthesized as described previously (Vejnar et al, 2016). Briefly, DNA templates for sgRNAs were synthesized by fill-in PCR using sgRNA primers containing the T7 promoter, the target sequence, partial sgRNA scaffold sequences, and a universal primer. sgRNA was transcribed in vitro

using HiScribe™ T7 Quick High Yield RNA Synthesis Kit (New England Biolabs). *cas9* mRNA was generated by in vitro transcription of XbaI linearized pT3TS-nCas9n plasmid (Addgene #46757) using the mMESSAGE mMACHINE™ T3 Transcription Kit (Invitrogen). About 30 pg each of sgRNA and 250 pg *cas9* mRNA were co-injected at the one-cell stage into TüLF wild-type embryos. Injected fish were raised to adulthood, and founders were identified by detecting deleted fragments by PCR. The founders were outcrossed with wild-type fish to generate an F1 generation. Each mutant allele was established from a single F1 fish. All experiments were performed using embryos produced by parents of the F2 generation onwards. Genotyping primers are listed in Table EV1.

### Antisense morpholino oligo injections into zebrafish

Knock down experiments were done by injecting 1–2 nL of antisense morpholino (MO) oligomers (Gene Tools, LLC) into one-cell stage embryos. Sequences of MOs are listed in Table EV1. Control embryos were injected with an equivalent amount of standard MO.

### Zebrafish pharmacological treatment

For the treatment of CCM mutants, the compounds MS37452 (Sigma), MS351 (Cayman Chemical), and BAY826 (Tocis) were used. Powdered compounds were dissolved in DMSO to produce 50 mM of stock solutions, and later diluted further in DMSO to generate 1000× solutions. Working solutions were prepared by diluting 1000× solutions in 1× Danieau's solution containing 0.0015% 1-phenyl-2-thiourea (PTU) (Sigma-Aldrich). At the 16-somite stage, approximately 20 dechorionated embryos were transferred into each well of a six-well plats and supplied with 4 mL of working solution. Embryos were then incubated until 54–56 hpf at 28.5 °C. Control embryos were treated with 0.1% DMSO/0.0015% PTU/1×Danieau's solution.

### Zebrafish tissue isolation, RNA extraction, and qRT-PCR

Heart tissues were extracted from 56 hpf embryos to harvest ~50 hearts/replicate as described (Lombardo et al, 2015). After extraction, pooled hearts were transferred into 100 μL of RNA lysis buffer from Quick RNA micro-prep (Zymo kit), and at least three biological replicates were collected for each condition. Tail and head samples were dissected from embryos using a syringe (10 embryos per replicate and four replicates for each condition), and subsequently, pooled samples were then transferred into TRIzol (Invitrogen). Total RNA was isolated according to the manufacturer's protocol and the corresponding cDNA was synthesized with the RevertAid H Minus First Strand cDNA Synthesis kit (Thermo Fisher Scientific). RT-qPCR experiments were performed as described previously (Renz et al, 2015) using either 2 ng (heart samples) or 20 ng (tail and head samples) of cDNA per technical replicate using the KAPA SYBR® Fast qPCR Kit (Sigma) on a qTower 3 qPCR thermal cycler (Analytik Jena). Cycle threshold (Ct) values were determined by the operating software of the device. *eif1b* was used as a housekeeping gene for normalization. Control sample values were normalized to 1.0, using the $2^{-\Delta\Delta Ct}$ method (Livak and Schmittgen, 2001). All primers are listed in Table EV1.

## Bulk-RNA-sequencing of zebrafish heart tissues

For RNA-sequencing, zebrafish heart tissues were extracted from wild-type, *ccm2^m201^* mutant or *ccm2^m20^;cbx7a^pbb62^* double mutants to harvest ~50 hearts/replicate as described (Lombardo et al, 2015). After extraction, pooled hearts were transferred into 100 µL of RNA lysis buffer from Quick RNA micro-prep (Zymo kit), and at least four biological replicates were collected for each condition. RNA was extracted according to the manufacturer's protocol and the integrity was assessed by the TapeStation system 2200 (Agilent). Only samples with a RIN score higher than 7 were used for further steps. cDNA libraries were synthesized from 30 ng of total RNA using NEBNext Poly(A) mRNA Magnetic Isolation Module (NEBNext® Ultra II Directional RNA Library Prep Kit for Illumina®, New England Biolabs). Samples were run on an Illumina NextSeq 550 sequencer using a High Output Kit v2.5 (75 cycles 400 M cluster, Illumina), and reads were aligned to the *Danio rerio* reference genome (GRCz10). Differential expression analysis and a heat map were analysed by DESeq2 (Galaxy Version 2.11.40.7 and heatmap2 package using the Galaxy platform (usegalaxy.eu).

## Whole-mount in situ hybridization

The *cbx7a* mRNA antisense template was generated by amplifying ~700 bp of the *cbx7a* coding sequence, with a reverse primer containing a T7 promoter overhang (Primers are listed in Table EV1). Antisense mRNA in situ probes were transcribed in vitro by the mMESSAGE mMACHINE™ T7 Transcription Kit (Invitrogen) together with the DIG RNA Labeling Kit (Roche). 54 hpf zebrafish embryos were fixed with 4% PFA overnight at 4 °C, and the whole-mount in situ hybridization was performed as previously described (Thisse and Thisse, 2008). Afterward, embryos were kept in absolute methanol until imaging. For imaging, specimens were first cleared in a mixture of benzyl benzoate and benzyl alcohol (ratio of 1:2) and mounted in Permount™ mounting medium. Images were recorded with 20x objectives on an Axioskop (Zeiss) with an EOS 5 D Mark III (Canon) camera and processed by Fiji software (Schindelin et al, 2012).

## Zebrafish whole-mount immunohistochemistry and imaging

Zebrafish whole-mount immunohistochemistry for Alcam was performed as previously described (Renz et al, 2015). The following antibodies were used: mouse anti-Zn-8/Alcam (1/25, Developmental Studies Hybridoma Bank), Alexa Fluor 546-conjugated goat anti-mouse (1/200, Thermo Fisher Scientific), Alexa Fluor 633-conjugated goat anti-mouse (1/200, Thermo Fisher Scientific). Briefly, embryos were fixed with 4% PFA overnight and blocked with PBST with 5% of NGS for 2 h and subsequently incubated overnight with the primary antibody diluted in PBST with 0.2% Triton X-100, 1% BSA, and 5% NGS. Later, embryos were washed in PBST and incubated overnight with a secondary antibody, followed by washing, mounting, and imaging. Specimens were mounted in SlowFade Gold (Thermo Fisher Scientific). Images were recorded on LSM 710 or LSM 880 confocal microscopes (Zeiss) and processed with Fiji.

## HUVEC cell culture and transfection

Pooled HUVECs were obtained from Lonza and grown in complete EGM-2 medium supplemented with 100 U/ml penicillin and 100 µg/ml streptomycin at 37 °C in a 5% $CO_2$ and humidified chamber according to the manufacturer's instructions. HUVECs ($1.5 \times 10^6$ cells) were transfected twice at 24 h intervals with 20 nM siRNA and 45 µl Lipofectamine RNAimax (Life Technologies, #13778-150) according to the manufacturer's instructions. For double transfections, 20 nM of each siRNA were used together with 45 µl Lipofectamine RNAimax. siRNA duplexes were from Dharmacon smartpool ONTARGET plus Perkin Elmer (CT: Non targeting 1: D-001810-01-50; CCM2: L-014728-01; CBX2: L-008357-01; CBX7: L-009561-01). HUVECs were regularly checked for mycoplasma contamination using the MycoAlert Mucoplasma Detection Kit from Lonza.

## HUVEC cell spreading and immunofluorescence

The day after the second round of transfection, transfected cells were trypsinized, treated with 1 mg/ml trypsin inhibitor (Sigma-Aldrich), and incubated in serum-free EBM-2 with 1% BSA for 30 min at 37 °C. Confluent HUVECs ($2 \times 10^5$ cells) were seeded in 24-well plates on slides coated with 10 µg/ml FN and incubated for 48 h in complete supplemented EGM-2 medium. Cells were immunostained as previously described (Lisowska et al, 2018; Manet et al, 2020). In brief, cells were fixed with 4% PFA, permeabilized with 0.2% Triton X-100, and incubated with anti-VE-cadherin antibody BV9 (Millipore) at a 1/200 dilution. After rinsing, coverslips were incubated with anti-mouse Alexa Fluor-conjugated secondary antibody (Invitrogen, #A32723) at 1/1000 and TRITC phalloidin (Sigma-Aldrich, #P-1951) at 1/2000 dilution. The coverslips were mounted in Mowiol/DAPI solution and imaged on an epifluorescent Axiomager microscope (Zeiss) with an AxioCamMRc camera.

## Permeability assay for HUVEC

The xCELLigence real-time cell analyzer (RTCA) system (Ozyme) was used to measure electrical impedance over time. Changes in the impedance of confluent endothelial cells reflect changes in barrier function (Twiss et al, 2012). Transfected HUVECs ($4 \times 10^4$ cells) were seeded at confluency in EGM-2 complete medium in 16 wells of E-plates (four wells per condition) previously coated with 10 µg/ml vitronectin and 50 µg/ml collagen I. After 4 h of adhesion, cells were starved in serum and cultured in basal EBM-2 containing 0.3% BSA for another 24 h. The cell index was normalized to the time of serum starvation to eliminate the signal due to cell spreading and adhering to the substrate. The normalized cell index at 24 h was plotted for the different conditions.

## Human-induced pluripotent stem cells (h-iPSC) and human-induced endothelial cell (h-iEC) cell culture

OdinCas9-hiPSCs (Lundin et al, 2020) were maintained in the Cellartis DEF-CS 500 Culture System (Takara Bio), according to manufacturer's instructions. All cell lines were cultured at 37 °C with 5% $CO_2$. Cell lines were authenticated by STR profiling and tested negative for mycoplasma. Prior to differentiation, cells were

adapted to the mTESR system (STEMCELL Technologies). H-iPSC cultured in DEF were dissociated into single cells with TrypLE Select (Thermo Fisher Scientific, #12563-029) and plated on Matrigel CLS354277 (Corning) at a density of 35000 cells/cm² in 50% mTeSR1 medium with 10 μM Y27632 and 50% DEF medium with GF3. After 24 h, the medium was changed to 75% mTeSR1 and 25% DEF, same ratio was used for media change after 48 h. Seventy-two hours after, cells were passaged again on Matrigel CLS354277 to 100% mTeSR system. They were cultured in mTeSR for at least two more passages before they were used for differentiation.

## h-iPSC *CCM2* cell line generation

Thirty-six hours prior to cell transfections, 40 k/cm² OdinCas9-hiPS cells were seeded into six-well plates. Sixteen hours prior, Cas9 was induced with 10 μg/ml of doxycycline for 1 h. A few hours before transfection, the media was changed to increase the likelihood of timing a proliferative log phase. The iPSCs cells were transfected with CRISPRMAX reagent using a 2.5:1 transfection reagent to sgRNA ratio and a reverse transfection protocol. For transfections, 40 k/cm² cells were seeded in a 96-well format directly onto prepared transfection complexes with 60 ng of sgRNA per well, as listed in Table EV1.

Cellartis® iPSC Single-Cell Cloning DEF-CS™ Culture Media Kit was used for single-cell cloning, and 0.5 cell/well were seeded on 384-well plates according to manufacturer´s instructions. When single-cell colonies were observed in wells, cells were expanded, and DNA samples were taken for next-generation sequencing (NGS). The rest of the cells were frozen down in freezing media (93%FBS, 7%DMSO). After NGS analysis three homozygous KO clones from each cell line were chosen for future work. Three clones where NGS results showed no edits were used as wild-type controls.

For karyotyping, cells were thawed in a DEF system and when they reached confluency, they were passaged to a T25 flask where they were cultured in mTeSR system (STEM CELL Technologies). When flasks were around 80% confluent, cells were prepared for karyotyping according to the protocol provided by Cell Guidance Systems and shipped to them for analysis. Karyotyping was evaluated by g-banding (Appendix Fig. S2F,G).

## Genomic DNA extractions and next-generation Amplicon sequencing of CCM2-depleted h-iPSC-derived ECs

Genomic DNA was extracted from cells using QuickExtract DNA extraction solution (Lucigen), according to the manual. Amplicons of interest were analyzed from genomic DNA samples on a NextSeq platform (Illumina). In brief, genomic sites of interest were amplified in a first round of PCR using primers that contained NGS forward and reverse adapters (Table EV1). The first PCR was setup using NEBNext Q5 Hot Start HiFi PCR Master Mix (New England Biolabs) in 15 μL reactions, with 0.5 μM of primers and 1.5 μL of genomic DNA as template. The PCR was carried out applying the following cycling conditions: 98 °C for 30 s; 30 cycles of [98 °C for 10 s; annealing temperature for each pair of primers for 20 s; and 72 °C for 20 s]; followed by a final 72 °C extension for 2 min. PCR products were purified using the HighPre PCR Clean-up System (MagBio Genomics), and correct PCR product size, and DNA concentration was analyzed on a Fragment Analyzer (Agilent). Unique Illumina indexes

were added to PCR products in a second round of PCR using KAPA HiFi Hotstart Ready Mix (Roche). Indexing primers were added in a second PCR step, and 1 ng of purified PCR product from the first PCR was used as a template in a 50 μL reaction volume. PCR was performed applying the following cycling conditions: 72 °C for 3 min; 98 °C for 30 s; then ten cycles of [98 °C for 10 s; 63 °C for 30 s; and 72 °C for 3 min]; followed by a final 72 °C extension for 5 min. Final PCR products were purified using the HighPre PCR Clean-up System (MagBio Genomics) and analyzed by Fragment analyzer (Agilent). Libraries were quantified using Qubit 4 Fluorometer (Life Technologies), pooled, and sequenced on a NextSeq instrument (Illumina).

## Differentiation to blood vessel endothelial cells

hiPSCs were differentiated into endothelial cells (hiECs) according to a previously published modETV2 protocol (Wang et al, 2020). After differentiation, hiECs were dissociated into single cells and sorted into CD31+ and CD31− cells using magnetic beads coated with anti-human CD31 antibodies (CD31 MicroBeads, Miltenyi Biotec). The purified CD31+ h-iECs were then expanded in culture on 6 well plates coated with Attachment Factor Solution (Cell applications). A culture medium for h-iECs was prepared by adding Endothelial Cell Growth medium 2 kit supplements into basal medium (PromoCell).

## Immunocytochemistry on hiECs

For immunohistochemistry, hiECs were seeded on a 96-well plate and fixed when they reached confluency with 4% formaldehyde, followed by permeabilization with 0.1% Triton X-100. The samples were blocked in 10% fetal bovine serum (FBS) for 1 h and incubated overnight at 4 °C with the Ve-Cadherin (Cell Signaling Technologies, #D87F2, dilution 1/400), or VWR (Sigma-Aldrich, #HPA001815, dilution 1/100) antibody in 5% FBS. After washing with PBS, cells were incubated with donkey anti-rabbit Alexa Fluor 488 (ThermoFisher, #A21206) in 1/1000 or Phalloidin 647 Plus (ThermoFisher, #A30107) in 1/400 with 5% FBS. DAPI (0.5 μg/ml) was used to stain the nucleus of the cells. The stained cells were analyzed with a Yokogawa CV7000S confocal microscope.

## Bulk RNA-seq CCM2-KO single-cell iPS clones

Total RNA was extracted from 50000–100000 cells using Agencourt RNAdvance Cell kit (Beckman Coulter) following the manufacturer's protocol and analyzed for quality and quantity on Fragment Analyzer 5300 system (Agilent). RNA-seq libraries were prepared using KAPA mRNA HyperPrep Kit (Roche) with an input of 50 ng total RNA and sequenced on NovaSeq6000 (Illumina) instrument with 51 bp paired-end setting with an average depth of 38mln clusters per sample.

## *CCM2*-KO iPSC-derived EC RNA-seq data processing and analysis

Libraries were assessed using FastQC (v0.11.7), Qualimap (v2.2.2c) (Okonechnikov et al, 2016) and SAMtools stats (v1.9). Alignment was performed using STAR (version 2.7.2b) (Dobin et al, 2013) with alignment against the human genome (GRCh38, Ensembl v100). Sequencing Quality control metrics were obtained using

Qualimap (v2.2.2c) (Okonechnikov et al, 2016) and summarized using MultiQC (v1.9). Trimming of adapters was performed using NGmerge (v0.3) (Gaspar, 2018). A human transcriptome index consisting of cDNA and ncRNA entries from Ensembl v100) was generated, and gene abundances were obtained using Salmon (v1.1.0) (Patro et al, 2017). The bioinformatics workflow was organized using the Nextflow workflow management system (v20.10) (DI Tommaso et al, 2017) and the Bioconda software management tool (Dale et al, 2018).

For differential gene expression analysis, DESeq2 (v1.26.0) was used with "ashr" (Stephens, 2017) for fold change shrinkage and a Wald test was used for pairwise comparisons, all in R (v4.0.2) (Love et al, 2014). Gene set enrichment analysis was performed with FGSEA (v1.16.0) (BioRxiv: https://doi.org/10.1101/060012) using Gene Ontology Biological Pathways collection (C5_BP of the MSigDB collection v7.5.1 (Liberzon et al, 2015)) and Reactome (C2_CP:REACTOME of the MSigDB collection v7.5.1 (Liberzon et al, 2015)).

## CCM2-KO iPS cell CUT&RUN data processing and analysis

Adapter sequences were trimmed and poor-quality reads were removed from the ends of reads using Trimmomatic (v0.39) (Bolger et al, 2014). Additionally, kseq (v20190822) was run to remove additional barcode sequences potentially missed through Trimmomatic processing(Zhu et al, 2019). Trimmed reads were aligned to the human genome (GRCH38, Gencode v 31) using Bowtie2 (v2.4.2) with the parameters "--very-sensitive-local" to assign multi-mapped reads to their best alignment and "--dovetail" to permit the alignment of mate pairs that extend past one another (Langmead and Salzberg, 2012). Reads were also aligned to the spike-in *E. coli* genome (K12-MG1655) using Bowtie2 with the additional parameters "--no-overlap" and "--no-dovetail". Aligned reads were filtered using SAMtools (v0.8.1) (Li et al, 2009) for a minimum quality score of 0. Spike-in normalization was performed using the procedure outlined in (Skene and Henikoff, 2017). Briefly, a scale factor was calculated for each sample based on the depth of the corresponding spike-in and Bedtools "genomecov" (v2.30.0) was used to scale target bam files and output a bedgraph file (Quinlan and Hall, 2010). For genome browser visualization, ucsc "bedGraphToBig-Wig" (v377) was used (Kent et al, 2010). For peak calling, macs2 "callpeak" (v2.2.7.1) and overlapping peaks across samples were merged using HOMER "mergePeaks" (v4.11) (Zhang et al, 2008; Heinz et al, 2010). Raw read quality was assessed using FastQC (version 0.11.9) (https://www.bioinformatics.babraham.ac.uk/projects/fastqc/), alignment metrics was assessed using picard (v2.25.0) (https://gatk.broadinstitute.org/hc/en-us/articles/360040507751-CollectAlignmentSummaryMetrics-Picard-) and SAMtools (v1.10), and binding enrichment quality metrics were generated through preseq (v3.1.2) (http://smithlabresearch.org/software/preseq/), deeptools (v3.5.0) (Ramírez et al, 2016), and Phantompeakqualtools (v1.2.2) (Landt et al, 2012). Quality metrics were summarized using MultiQC (v1.10.1) (Ewels et al, 2016). Differentially bound peaks were identified using DiffBind (v3.4.3) (Ross-Innes et al, 2012) and over-representation gene set analysis were performed using clusterProfiler (v4.2.0) (Wu et al, 2021) against Gene Ontology Biological Pathways with org.Hs.erg.db (v3.14.0) annotations. Data processing was managed using Snakemake (v5.26.1) (Köster et al, 2021).

## Human CCM brain biopsies

Paraffin-embedded biopsies from patients with familial CCM were obtained from the Alliance to Cure Cavernous Malformation CCM Biobank through a Material Transfer Agreement. The patients have given their consent for the use of biopsies for research studies. Collection and use of the samples in research was approved by the Swedish Ethical Review Authority (EPM; 2019-04715 and 2019-06374). The ethical authorities in Sweden follow the Helsinki Declaration 2013 and the Department of Health and Human Services Belmont Report.

## Immunohistochemistry of human CCM biopsies and imaging

Immunohistochemical stainings were performed using fully automated protocols (DAKO Autostainer Link48) and Envision FLEX, a high pH detection kit from DAKO #K8000. The following antibodies were used: CBX7 (Sigma-Aldrich, #HPA056480, diluted 1/200), CD34 (Dako, #IR632). Images were recorded on Axioscan Z.1 (Zeiss) and processed with Fiji.

## Mouse models

All the procedures with mice were performed in agreement with the Institutional Animal Care and Use Committee (IACUC) of IFOM ETS, AIRC Institute of Molecular Oncology, in compliance with the guidelines established in the Principles of Laboratory Animal Care (Directive 86/609/EEC) and as approved by the Italian Ministry of Health. The animals were housed under standard conditions and were fed with a regular chow diet ad libitum. Cdh5(PAC)-Cre-ER$^{T2}$/CCM3$^{flox/flox}$ mice (Bravi et al, 2015; Wang et al, 2010) were used for the experiments performed in this project. These mouse lines (here named CCM3iECKO) derived from the breeding of CCM3$^{flox/flox}$ mice with the vascular endothelial cadherin-Cre-ERT2 (Cdh5(PAC)-Cre-ER$^{T2}$) mice. These mice express the *Cre*-recombinase fused to the mutated form of the human estrogen receptor (ER$^{T2}$). ER$^{T2}$ is activated by tamoxifen, at low levels, but not by the endogenous estrogen. This fusion protein is expressed under the vascular endothelial cadherin (Cdh5) promoter, therefore only in endothelial cells. Upon tamoxifen injection [or its active metabolite 4-hydroxy-tamoxifen (4-OHT)], the fusion protein Cre-recombinase-ER$^{T2}$ is released from heat-shock proteins (Hsp) and moves to the nucleus where Cre-recombinase acts over loxP sites. The brains of wild-type and CCM3$^{iECKO}$ mice were collected at postnatal day 8 (P8). The gender of the mice was randomly maintained with a 1:1 composition of males and females.

## Endothelial-specific recombination of mouse lines with tamoxifen

CCM3iECKO mice were treated with tamoxifen (Sigma-Aldrich) to specifically induce recombination of floxed genes in endothelial cells. Tamoxifen was first dissolved in pre-warmed (37–40 °C) ethanol to a final concentration of 100 mg/mL. Then, pre-warmed corn oil was slowly added, to a final concentration of 10 mg/mL. Tamoxifen (10 mg/mL) was aliquoted and stored in the dark at −20 °C. On the day of injection, an aliquot of tamoxifen was diluted to a final concentration of 2 mg/mL in corn oil and 50 mL of

the solution was injected intragastric into each mouse at 1 day after birth (P1).

## Mouse genotyping

To prepare DNA for genotyping, tails or fingertips from mice were digested in lysis buffer (10% buffer Gittschier, TX-100, 0.5 mg/ml Proteinase K, H2O) overnight at 56 °C shaking. Lysed tissues were then heated at 95 °C for 5 min before being genotyped. The primers used for mouse genotyping are listed in Table EV1.

## Mouse treatment with MS37452

Starting from the day after the tamoxifen injection, Cre-positive CCM3iECKO mice received daily an intragastric injection of MS37452 (2 mg/g body weight). The drug was first dissolved in DMSO and then diluted in 50 mL of corn oil. The control mice were treated with the same amount of DMSO, dissolved in 50 mL of corn oil. The brains of CCM3iECKO mice were collected at postnatal day 8 (P8), and fixed in 4% paraformaldehyde (PFA; Sigma-Aldrich) at 4 °C overnight.

## Immunofluorescence on murine brain tissues

Fixed brains were embedded in 4% low-melting-point agarose and sectioned (100 mm) along the sagittal axis using a vibratome (1000 Plus, The Vibratome Company, St. Louis, MO, US). Brain sections were stained as floating samples in 6- or 12-well plates. The blocking of non-specific binding sites was performed for 1 h at room temperature (RT) in 0.3% Triton X-100, 5% donkey serum, and 1% bovine serum albumin (BSA) in phosphate-buffered Saline (PBS). The samples were incubated overnight at 4 °C with the primary antibodies diluted in the same solution that was used for blocking. Several washes in 0.1% Triton X-100 in PBS were performed before the incubation with secondary antibodies for 4 h at RT in 0.3% Triton X-100, 1% BSA in PBS. After secondary antibody incubation further washes in PBS were performed, followed by a post-fixation step with 4% PFA for 5 min at RT. Further washes in PBS1X were done after post-fixation step. The brain sections were mounted in Vibrant Vectashield with 4′,6-diamidino-2-phenylindole (DAPI).

For mouse brains, vessels were stained with biotinylated Isolectin B4 (Vector Laboratories, #B-1205, dilution 1/200) and then with streptavidin conjugated with Alexa Fluor 647 (Invitrogen, S#21374, dilution 1/400). The following secondary antibodies were used (produced in either donkey or goat and targeted against the appropriate species) with conjugation with Alexa Fluor 488 or 647 (Jackson Laboratories, #AB_2340846; #AB_2338902, dilution 1/400).

## Quantification and statistical analysis of murine vascular lesions

For lesions quantification, 100-mm-thick vibratome sections of brain were prepared and stained for biotinylated Isolectin B4. For each brain, at least five sagittal sections of the cerebellum were cut, and z-stacks were acquired with a 10X objective using a confocal microscope, SP8 DLS (Leica). Lesions were subsequently quantified as described previously (Maderna et al, 2022). For the downstream analysis, the number and total area of lesions were quantified.

### The paper explained

#### Problem

Cerebral cavernous malformation (CCM) is a vascular disease caused by mutations in the *CCM1/KRIT1*, *CCM2*, or *CCM3/PDCD10* genes. These mutations trigger the transcription factor Krüppel-like factor 2 (KLF2) to activate pathological changes in endothelial cell gene expression. Identifying the genes that become activated by KLF2 in the pathology of CCM is of great biomedical interest.

#### Results

Our research reveals that the Chromobox Protein Homolog 7 (CBX7), which modifies genome organization, also changes gene activation by KLF2. Higher levels of CBX7 were found in endothelial cells of patients with CCM and in mouse and zebrafish pre-clinical CCM disease models, compared with healthy subjects. The pathological CCM phenotypes in zebrafish embryos, CCM2-deficient human umbilical vein endothelial cells (HUVECs), and in a pre-clinical CCM3 mouse disease model, were suppressed by genetic or pharmacological CBX7 inhibition. Furthermore, both KLF2 and blood flow regulated the expression of CBX7 in zebrafish. In turn, CBX7 was involved in the activation of KLF2 target genes, including genes related to endothelial-to-mesenchymal transition (endoMT), *WNT9*, and *TEK*.

#### Impact

Our findings show that CBX7 directs KLF2 towards pathological target genes. This discovery offers potential therapeutic strategies, as targeting CBX7 may suppress the harmful effects of CCM mutations in patients and improve their condition.

Subsequently, lesions were classified as small or large according to a cut-off set at 5000 µm². Statistical analyses were performed with GraphPad Prism 5. The statistical testing methods used for each experiment and $p$ values calculations are indicated in the figure legends. All indicated $p$ values are two-tailed and significance was defined as $p < 0.05$.

## Laser-captured human surgically resected CCM lesions and normal brain capillaries

Ten surgically resected human CCM lesions (6 sporadic/solitary and 4 familial/multifocal) were embedded in optimal cutting temperature (OCT), snap frozen, and stored at −80 °C (Koskimäki et al, 2019; Lyne et al, 2019). Normal brain tissue of 3 patients free of neurological disease was sampled during autopsy, fixed in formalin, and embedded in paraffin blocks (FFPE). Five-µm sections of tissue brain sample were first mounted onto Leica glass slides (Leica Biosystems Inc., Buffalo Grove, Illinois, USA) and stained with either HistoGene (Applied Biosystems, California, USA) or Paradise stain (Applied Biosystems) for FFPE tissue. The endothelial layer, including adjacent luminal and extraluminal elements, from CCMs and normal brain capillaries were then collected at 40x of magnification using a Leica LMD 6500 laser capture microdissection system (Leica Biosystems, Wetzlar, Germany), and stored at −80 °C (Koskimäki et al, 2019; Lyne et al, 2019). RNA was extracted using an RNA isolation kit (RNeasy® Micro Kit, Qiagen, Hilden, Germany). cDNA libraries were generated using commercially low-input strand specific RNA-Seq kits (Clontech, California, USA), and sequenced on the Illumina HiSeq4000 platform using single-end 50 basepair (bp) reads. The

quality of raw sequencing reads was assessed by FastQC v0.11.2 (available at: http://www.bioinformatics.babraham.ac.uk/projects/fastqc/). The post-alignment quality control was evaluated using RSeQC, and Picard tools v1.117 (Wang et al, 2012). Reads were mapped into the GENCODE human genome model (GRCh38 V28) using STAR(Dobin et al, 2013). Differentially-expressed genes were identified with DESeq2, using an additive model to correct for batch effect when needed (Love et al, 2014).

## Data availability

Primary dataset for zebrafish RNA-sequencing experiment has been deposited in Gene Expression Omnibus (GEO) database with accession number of GSE277234. Raw sequencing data from human patient RNA-sequencing is available with accession number of GSE130176. Due to Swedish law, the patient consent, and the risk that the sequencing data contains personally-identifiable information and hereditary mutations, raw sequencing data from RNA- and CUT&RUN experiments of iPSC-derived endothelial cells were not deposited in a public repository.

The source data of this paper are collected in the following database record: biostudies:S-SCDT-10_1038-S44321-024-00152-9.

## Peer review information

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

## Acknowledgements

We would like to thank all members of the Seyfried group for critical reading and comments on the manuscript. Thanks to O. Baumann for his support of confocal microscopy and T. Kollenkirchen, A. Kühnel, A. Hubig, B. Wuntke, and M. Novatchkova for technical support. We are indebted to Ralf Adams for providing Cdh5(PAC)-CreERT2 mice. A particular thank goes to the Alliance to Cure Cavernous Malformation for providing samples from the human CCM biopsy bank. We thank Forskning, utveckling och utbildning (FoUU) Service, Uppsala University Hospital for support of immunohistological stainings of human CCM biopsies. SA-S was generously supported by SFB958, Deutsche Forschungsgemeinschaft (DFG) projects SE2016/7-3, SE2016/10-1, SE2016/13-1, the Leducq Transatlantic Network of Excellence "21CVD03 - ReVAMP", and the *Marie Skłodowska-Curie* Innovative Training Network "V.A.Cure". (*ITN*). The LSM 880 AiryScan was funded by DFG grant INST 336/114-1 FUGG. EF was supported by Espoir Isère contre le cancer. CP was funded by ANRT. IAA and RG were supported by US NIH/NINDS grant # P01NS092521, and JK was supported by the Sigrid Juselius Foundation. MH and SS-M were supported by the DFG (CRC1348B08). MP was generously supported by AIRC grant IG2019-ID 23826; Fondazione AIRC under 5 per mille 2019, ID 22759 program; Telethon GGP19202; and Ministero dell'Istruzione, dell'Università e della Ricerca award PNRR M4C2-Investimento 1.4-CN00000041 – NextGenerationEU.

## Author contributions

**Van-Cuong Pham**: Conceptualization; Resources; Software; Formal analysis; Investigation; Visualization; Methodology; Writing—original draft; Writing—review and editing. **Claudia Jasmin Rödel**: Conceptualization; Resources; Data curation; Software; Formal analysis; Supervision; Investigation; Visualization; Methodology; Writing—original draft; Writing—review and editing. **Mariaelena Valentino**: Data curation; Formal analysis; Validation; Investigation; Visualization; Methodology; Writing—original draft; Writing—review and editing. **Matteo Malinverno**: Data curation; Formal analysis; Investigation; Methodology. **Alessio Paolini**: Resources; Investigation; Visualization; Methodology. **Juliane Münch**: Resources; Data curation; Software; Formal analysis; Supervision; Validation; Investigation; Methodology. **Candice Pasquier**: Data curation; Formal analysis; Investigation; Methodology. **Favour C Onyeogaziri**: Resources; Data curation; Formal analysis; Investigation; Methodology. **Bojana Lazovic**: Resources; Data curation; Formal analysis; Investigation; Methodology. **Romuald Girard**: Resources; Software; Validation; Investigation; Methodology. **Janne Koskimäki**: Resources; Data curation; Software; Formal analysis; Investigation; Methodology. **Melina Hußmann**: Resources; Investigation; Methodology. **Benjamin Keith**: Resources; Software; Formal analysis; Investigation; Methodology. **Daniel Jachimowicz**: Resources; Data curation; Software; Formal analysis; Investigation; Visualization; Methodology. **Franziska Kohl**: Resources; Software; Formal analysis; Investigation; Methodology. **Astrid Hagelkruys**: Resources; Investigation; Methodology. **Josef M Penninger**: Resources; Investigation; Project administration. **Stefan Schulte-Merker**: Resources; Supervision; Investigation; Methodology; Project administration; Writing—review and editing. **Issam A Awad**: Resources; Data curation; Supervision; Project administration. **Ryan Hicks**: Resources; Formal analysis; Supervision; Investigation; Methodology; Project administration. **Peetra U Magnusson**: Conceptualization; Resources; Data curation; Supervision; Validation; Investigation; Methodology; Project administration. **Eva Faurobert**: Resources; Data curation; Formal analysis; Supervision; Validation; Investigation; Methodology; Project administration. **Massimiliano Pagani**: Resources; Data curation; Formal analysis; Supervision; Validation; Investigation; Methodology; Writing—original draft; Project administration; Writing—review and editing. **Salim Abdelilah-Seyfried**: Conceptualization; Resources; Data curation; Supervision; Funding acquisition; Validation; Investigation; Writing—original draft; Project administration; Writing—review and editing.

Source data underlying figure panels in this paper may have individual authorship assigned. Where available, figure panel/source data authorship is listed in the following database record: biostudies:S-SCDT-10_1038-S44321-024-00152-9.

## Funding

## Disclosure and competing interests statement

SA-S holds a European patent (EP4 154 876) related to this study.

# Expanded View Figures

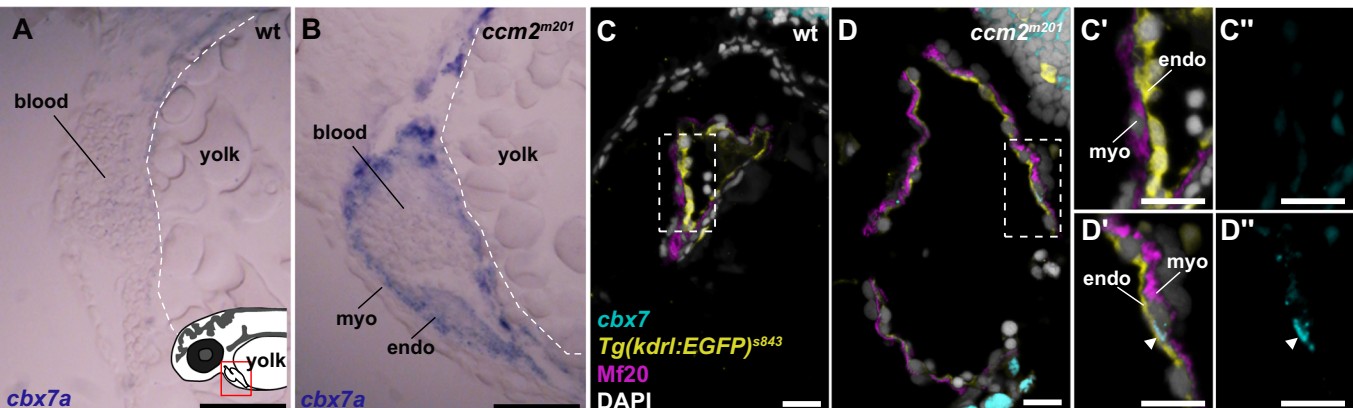

**Figure EV1.** *cbx7a* **is upregulated in the zebrafish endocardium.**

(**A, B**) Shown are sagittal sections of whole-mount in situ hybridizations of zebrafish embryos at 56 hpf in wild-type (wt) (**A**) and *ccm2^{m201}* mutants (**B**), revealing elevated expression levels of *cbx7a* mRNA in *ccm2^{m201}* mutants throughout the entire endocardium. (**C, D**) Shown are confocal optical sections of whole-mount fluorescent in situ hybridization for *cbx7a* transcripts and immunohistological co-staining for myocardial marker Mf20. The endocardium is marked by *Tg(kdrl:EGFP)^{s843}* and nuclei are marked by DAPI. Transcripts of *cbx7a* are detected only in the endocardium of *ccm2^{m201}* mutants (**D, D′,D″**, arrowheads indicate the location of *cbx7a* transcripts), whereas wild-type embryos lack signals for *cbx7a* (**C, C′, C″**). Scale bars are (**A, B**) 100 μm; (**C, D**) 20 μm; (**C′, D″**) 10 μm. Source data are available online for this figure.

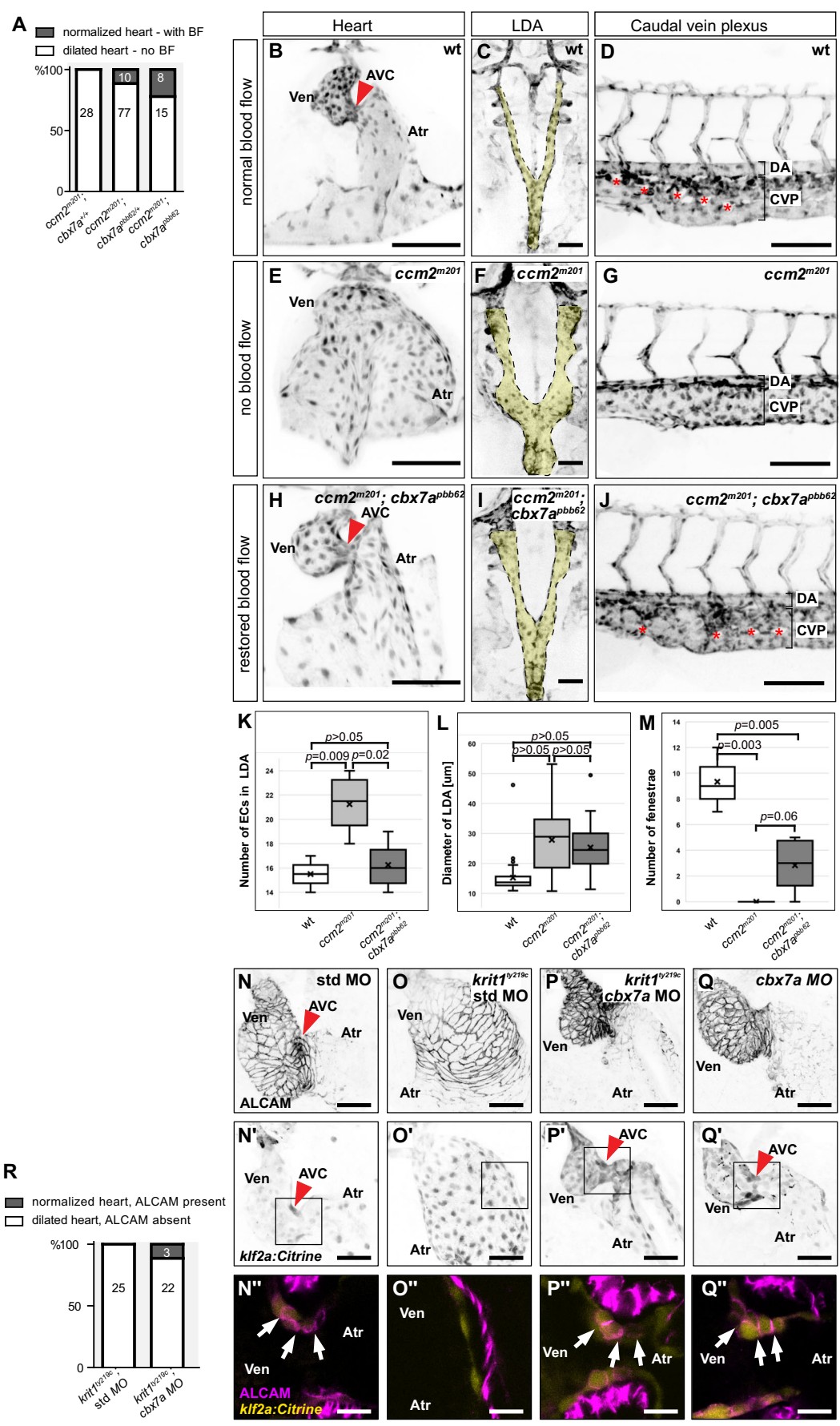

◀ **Figure EV2. CCM mutant cardiovascular defects are suppressed in zebrafisch embryos upon depletion of Cbx7a.**

(A) Shown are quantifications of the share of $ccm2^{m201}$ mutant embryos in which cardiovascular phenotypes are normalized, which is never seen in $ccm2^{m201}$ mutants. Normalization was assessed by the presence of a constricted atrioventricular canal (AVC) and blood flow (BF). (B–J) Shown are maximum projections of confocal microscopic z-stacks with comparisons of cardiovascular phenotypes in wild-type (wt), $ccm2^{m201}$ mutants, and $ccm2^{m201};cbx7a^{pbb62}$ double mutants. The morphology of the heart is normalized upon the loss of $cbx7a$ in $ccm2^{m201}$ mutants (B, E, H). The lateral dorsal aorta is dilated in $ccm2^{m201}$ mutants, and $ccm2^{m201};cbx7a^{pbb62}$ double mutants (C, F, I; highlighted in yellow). The caudal vein plexus (CVP) is dilated and fused into a single tube in $ccm2^{m201}$ mutants. Asterisks indicate spaces (fenestrae) within the caudal vein plexus, which do not form in $ccm2^{m201}$ mutants but are present in wild-type and $ccm2^{m201};cbx7a^{pbb62}$ double mutants (D, G, J). (K–M) Quantifications of LDA endothelial cell numbers ($n = 4$ for each condition) and diameter ($n = 4$ for each condition), and number of fenestrae in caudal vein plexus (wild-type, $n = 3$; $ccm2^{m201}$, $n = 3$; $ccm2^{m201};cbx7a^{pbb62}$ double mutants, $n = 6$) Standard boxplots show median and quartiles with minima and maxima indicated by whiskers (statistical testing is based on pairwise student's $T$-test). The x represents the mean value. Endothelial cell numbers in the lateral dorsal aorta) are normalized (K), albeit vessel diameter remains mostly unchanged (L). The caudal vein plexus (CVP) is normalized upon loss of $cbx7a$ in $ccm2^{m201}$ mutants (M). (N–Q) Functional rescue of endocardial phenotypes through genetic depletion of Cbx7a in zebrafish $krit1^{ty219c}$ mutant embryos. Shown are maximum projections of confocal image z-stacks of zebrafish hearts at 56 hpf with myocardium marked with ALCAM (N–Q) and the endocardium being marked by $Tg(klf2a:Citrine)^{mu107}$ expression (N'–Q'). The expression of ALCAM in endocardial AVC cells (N''; arrows) is lost in $krit1^{ty219c}$ mutants (O''). The depletion of $cbx7a$ using an antisense oligo morpholino ($cbx7a$ MO) rescues cardiac morphology in $krit1^{ty219c}$ mutant hearts. In 3 of 25 embryos, the AVC region is restored (P', red arrowhead), and ALCAM expression at the endocardial AVC is restored (P''; arrows; R). Injection of a control morpholino does not have any effects on overall morphology or ALCAM expression (N, N', N''). Knockdown of $cbx7a$ alone does not affect AVC formation and ALCAM expression (Q, Q', Q''). AVC atrioventricular canal, Ven ventricle, Atr atrium. Statistical testing is based on unpaired student's $T$-test between (s.e.m. bars indicated). Red arrowheads indicate AVC while white arrows indicate the presence of ALCAM expression in AVC endocardial cells. AVC atrioventricular canal, CVP caudal vein plexus, Ven ventricle, Atr atrium. Scale bars are (B, D, E, G, H, J) 100 μm; (N–Q, N'–Q') 50 μm; (C, F, I, N''–Q'') 10 μm. Source data are available online for this figure.

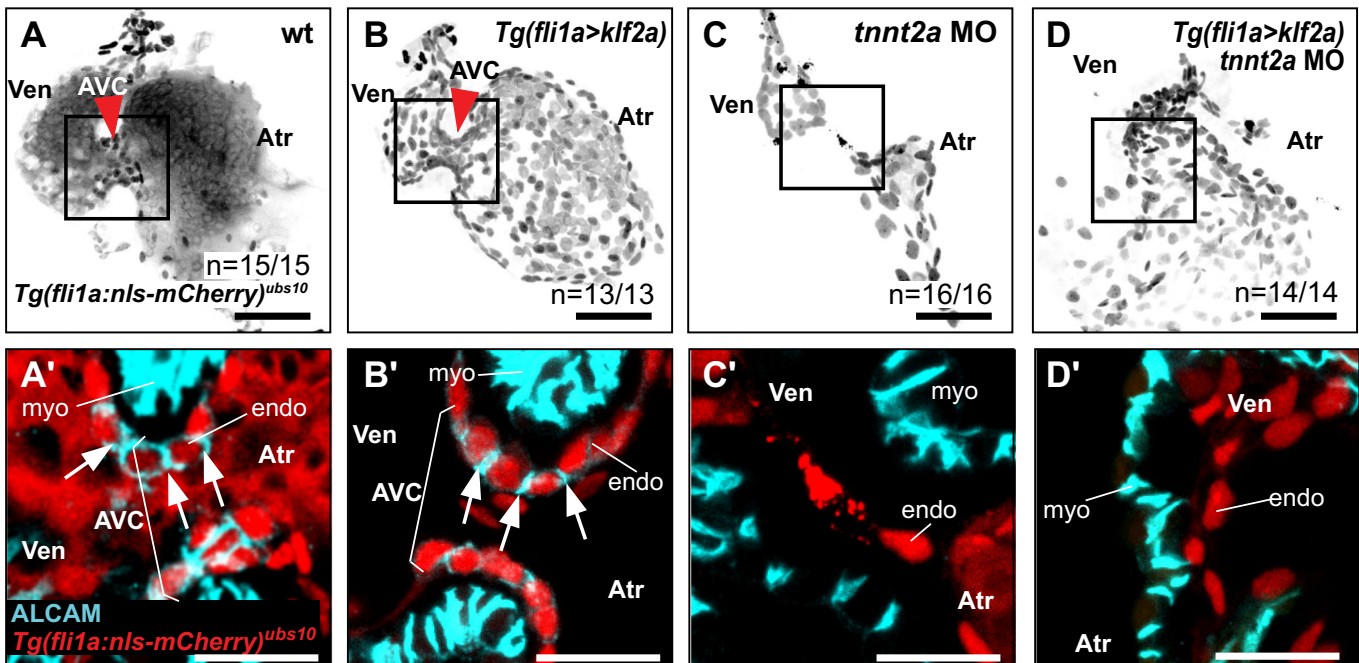

**Figure EV3. Concomitant loss of blood flow and endothelial overexpression Klf2a causes CCM-like cardiac phenotypes in zebrafish embryos.**

(A–D) Shown are confocal z-scan projections of zebrafish embryonic endocardium marked by expression of *Tg(fli1a:nls-mCherry)*[ubs10] at 48 hpf of wild-type (**A**, **A'**), *Tg(fli1a:Gal4FF)*[ubs3]; *Tg(UAS:klf2a)*[ig1] double transgenic embryos [*Tg(fli1a>klf2a)*]. (**B**, **B'**) An antisense morpholino oligo against *tnnt2a* injected into wild-type (**C**, **C'**) and *Tg(fli1a>klf2a)* (**D**, **D'**). Inserts (**A'–D'**) provide magnifications based on projections of fewer z-scan section planes of the atrioventricular canal (AVC) region. In comparison to wild-type (**A**, **A'**), *Tg(fli1a>klf2a)* hearts show a ballooning of the atrium (**B**), while the overall morphology of the AVC appears normal, with ALCAM expression intact at the endocardial cells of the AVC (**B**, red arrow; **B'**, arrowheads). The MO-mediated depletion of *tnnt2a* in wild-type leads to a severe reduction in cardiac size (**C**), and endocardial cells of the AVC fail to express ALCAM (**C'**). Knock-down of *tnnt2a* in *Tg(fli1a>klf2a)* augments the cardiac ballooning, leading to loss of the AVC constriction (**D**) and loss of expression of ALCAM in endocardial AVC cells (**D'**). Numbers (*n*) indicate observed phenotypes in different conditions. Red arrowheads indicate AVC while white arrows indicate the presence of ALCAM expression in AVC endocardial cells. AVC atrioventricular canal, endo endocardium, myo myocardium, OFT outflow tract, Ven ventricle, Atr atrium. Scale bars are (**A–D**) 50 μm, (**A'–D'**) 20 μm. Source data are available online for this figure.

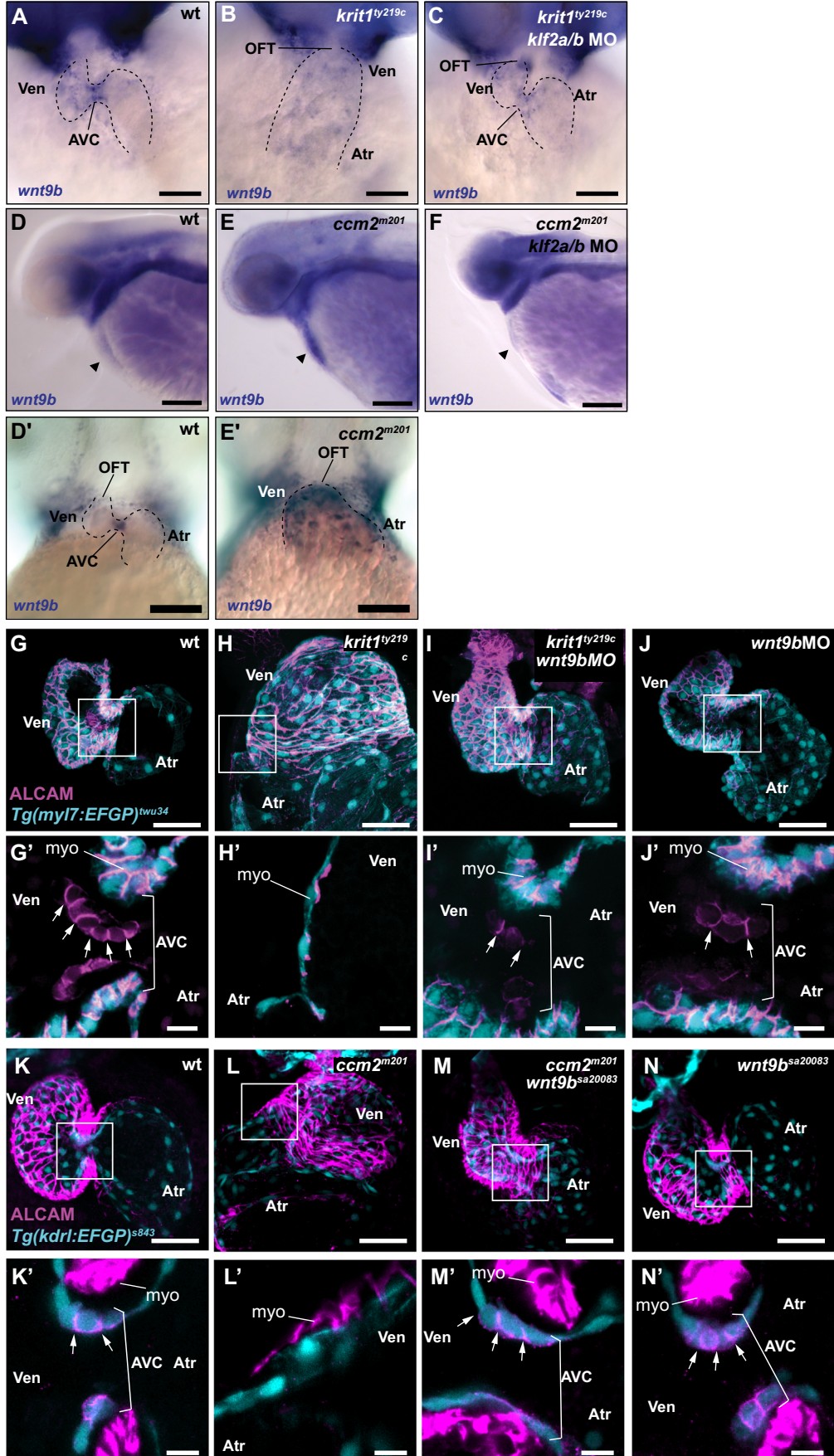

**Figure EV4.** The upregulation of *wnt9b* expression in *ccm* zebrafish mutants involves Klf2 and contributes to CCM phenotypes.

(A–F) Representative images of whole-mount in situ hybridizations against *wnt9b* mRNA in the zebrafish heart (outlined by dotted lines) at 48 hpf (D–F, lateral views). *wnt9b* mRNA expression is restricted to the AVC region in wild-type embryos (A,D,D') and elevated throughout the entire heart in *ccm2*$^{m201}$ and *krit1*$^{ty219c}$ mutants (B, E, E'). (C, F) Upon knock-down of *klf2a* and *klf2b* in *ccm2*$^{m201}$ and *krit1*$^{ty219c}$ mutants, expression of *wnt9b* is restricted to the AVC region. (G–N) Shown are confocal z-scan projections of zebrafish embryonic hearts at 48 hpf with inserts (G'–N') providing magnifications based on projections of fewer z-scan section planes of the atrioventricular canal (AVC) region. The myocardium is marked by *Tg(myl7:EGFP)*$^{twu34}$ expression and ALCAM (G–J, G'–J') or only ALCAM (K–N; K'–N'). In comparison to the heart morphology in wild-type (G, G', arrows indicate ALCAM expression in endocardial cells of the AVC $n = 10/10$ hearts), *krit1*$^{ty219c}$ mutant hearts exhibit a ballooning morphology (H, $n = 4/4$ hearts) and lack ALCAM expression in endocardial cells of the AVC (H'). The morpholino-mediated depletion of *wnt9b* in *krit1*$^{ty219c}$ mutants normalizes the heart morphology with a constriction at the AVC (I) and ALCAM expression in AVC endocardial cells (I'; arrows). The morpholino-mediated knock-down of *wnt9b* in wild-type does not affect AVC formation and endocardial ALCAM expression (J, J', $n = 3/3$ hearts). In *ccm2*$^{m201}$ mutant hearts, the cardiac ballooning morphology (L) and lack of ALCAM expression in endocardial cells of the AVC (L') is restored only in embryos that are also mutant for *wnt9b*$^{sa20083}$ (M, M', $n = 5/14$ *ccm2*$^{m201}$ mutant hearts) with a normalized heart morphology and a constriction at the AVC. In comparison, *wnt9b*$^{sa20083}$ mutants do not exhibit defects in cardiac morphology or ALCAM expression (N, N', $n = 4/4$ hearts). Black arrowheads indicate the location of the heart. White arrows indicate the presence of ALCAM expression in AVC endocardial cells. AVC atrioventricular canal, myo myocardium, Ven ventricle, Atr atrium. Scale bars are (A–F) 100 μm, (G–J, K–N) 50 μm, (G'–J', K'–N') 10 μm. Source data are available online for this figure.

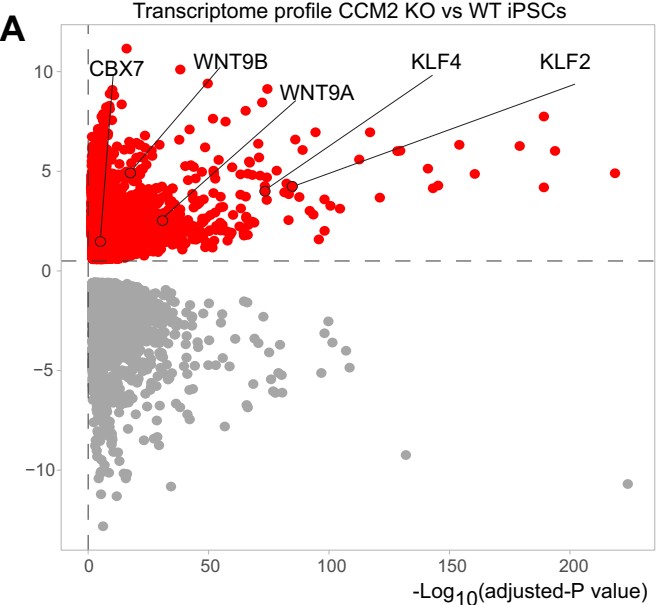

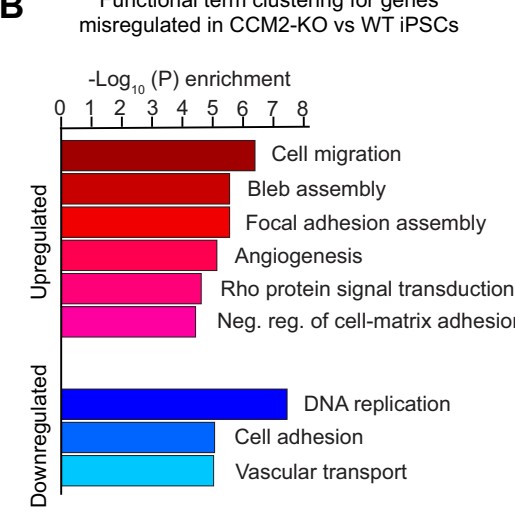

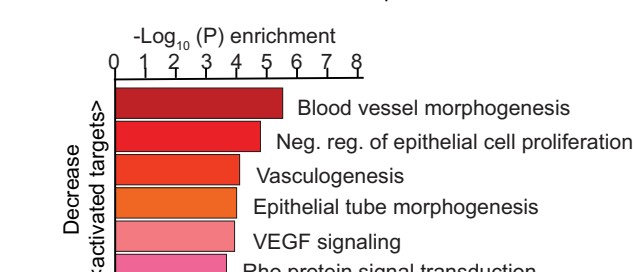

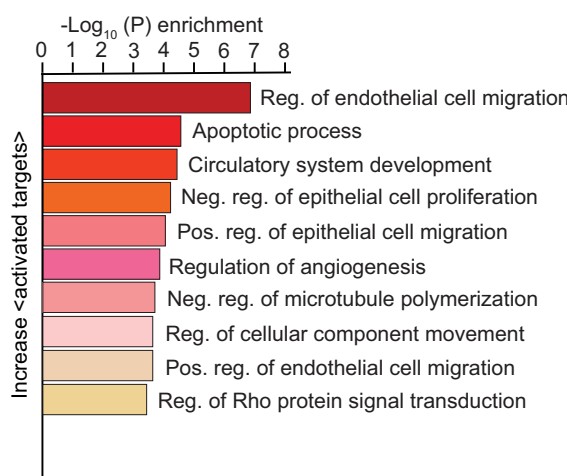

**Figure EV5. Transcriptional and epigenetic regulation in *CCM2* knock out (KO) iPSC-derived ECs.**

(A) Volcano plot of RNA-seq data of differentially-expressed genes (calculated by Wald test under DESeq2 package (v1.26.0), adjusted *p* value <0.05 and fold change >1) in CCM2-depleted iPCS-derived ECs compared to wild-type. Indicated is the upregulation of the CCM hallmark marker genes *KLF2* and *KLF4* as well as *CBX7*, *WNT9A*, and *WNT9B*. (B) Functional clustering of gene ontology terms for genes misregulated in *CCM2*-KO as compared to WT. (C, D) Functional clustering of gene ontology terms for inactivating H3K27me3 mark target genes (both increased and decreased targets) (C) and increased activating H3K4me3 mark gene targets (D) in CUT&RUN-seq data from *CCM2*-KO and WT (differentially bound peaks were calculated by Wald test under DESeq2 associated with Diffbind (v.3.4.3) package). Source data are available online for this figure.

