## [Peer Review File · EMBO Molecular Medicine]

Epigenetic regulation by Polycomb Repressive Complex 1 promotes Cerebral Cavernous Malformations

Van-Cuong Pham, Claudia Rödel, Mariaelena Valentino, Matteo Malinverno, Alessio Paolini, Juliane Münch, Candice Pasquier, Favour Onyeogarizi, Bojana Lazovic, Romuald Girard, Janne Koskimäki, Melina Hußmann, Benjamin Keith, Daniel Jachimowicz, Franziska Kohl, Astrid Hagelkruys, Josef Penninger, Stefan Schulte-Merker, Issam Awad, Ryan Hicks, Peetra Magnusson, Eva Faurobert, Massimiliano Pagani, and Salim Abdelilah-Seyfried

Corresponding author(s): Salim Abdelilah-Seyfried (sseyfried@uni-potsdam.de) , Massimiliano Pagani (massimiliano.pagani@ifom.eu)

Review Timeline:

Submission Date:	19th Feb 24
Editorial Decision:	20th Mar 24
Revision Received:	14th Aug 24
Editorial Decision:	10th Sep 24
Revision Received:	19th Sep 24
Accepted:	27th Sep 24

Editor: Lise Roth

Transaction Report:

20th Mar 2024

Dear Dr. Seyfried,

Thank you for the submission of your manuscript to EMBO Molecular Medicine. We have now received feedback from the three reviewers who agreed to evaluate your manuscript. As you will see from the reports below, the reviewers acknowledge the interest of the study, but also raise several concerns that should be addressed in a revision of this manuscript.

EMBO Molecular Medicine encourages a single round of revision only and therefore, acceptance or rejection will depend on the completeness of your responses included in the next, final version of the manuscript. For this reason, and to save you from any frustrations in the end, I would strongly advise against returning an incomplete revision. In particular, it will be important to emphasize the relevance of your findings to human disease.

We are expecting your revised manuscript within three months, if you anticipate any delay, please contact us.

We require:

- 1) A .docx formatted version of the manuscript text (including legends for main figures, EV figures and tables). Please make sure that the changes are highlighted to be clearly visible.
- 2) Individual production quality figure files as .eps, .tif, .jpg (one file per figure). For guidance, download the 'Figure Guide PDF' (<https://www.embopress.org/page/journal/17574684/authorguide#figureformat>).
- 3) At EMBO Press we ask authors to provide source data for the main figures. Our source data coordinator will contact you to discuss which figure panels we would need source data for and will also provide you with helpful tips on how to upload and organize the files.
- 4) A .docx formatted letter INCLUDING the reviewers' reports and your detailed point-by-point responses to their comments. As part of the EMBO Press transparent editorial process, the point-by-point response is part of the Review Process File (RPF), which will be published alongside your paper.
- 5) A complete author checklist, which you can download from our author guidelines (<https://www.embopress.org/page/journal/17574684/authorguide#submissionofrevisions>). Please insert information in the checklist that is also reflected in the manuscript. The completed author checklist will also be part of the RPF.
- 6) Please note that all corresponding authors are required to supply an ORCID ID for their name upon submission of a revised manuscript. An ORCID identifier is currently missing for Massimiliano Pagani.
- 7) It is mandatory to include a 'Data Availability' section after the Materials and Methods. Before submitting your revision, primary datasets produced in this study need to be deposited in an appropriate public database, and the accession numbers and database listed under 'Data Availability'. Please remember to provide a reviewer password if the datasets are not yet public (see <https://www.embopress.org/page/journal/17574684/authorguide#dataavailability>). In case you have no data that requires deposition in a public database, please state so in this section. Note that the Data Availability Section is restricted to new primary data that are part of this study.
- 8) For data quantification: please specify the name of the statistical test used to generate error bars and P values, the number (n) of independent experiments (specify technical or biological replicates) underlying each data point and the test used to calculate p-values in each figure legend. The figure legends should contain a basic description of n, P and the test applied. Graphs must include a description of the bars and the error bars (s.d., s.e.m.). Please provide exact p values.
- 9) Our journal encourages inclusion of *data citations in the reference list* to directly cite datasets that were re-used and obtained from public databases. Data citations in the article text are distinct from normal bibliographical citations and should directly link to the database records from which the data can be accessed. In the main text, data citations are formatted as follows: "Data ref: Smith et al, 2001" or "Data ref: NCBI Sequence Read Archive PRJNA342805, 2017". In the Reference list, data citations must be labeled with "[DATASET]". A data reference must provide the database name, accession

number/identifiers and a resolvable link to the landing page from which the data can be accessed at the end of the reference. Further instructions are available at .

13) Author contributions: CRediT has replaced the traditional author contributions section because it offers a systematic machine readable author contributions format that allows for more effective research assessment. Please remove the Authors Contributions from the manuscript and use the free text boxes beneath each contributing author's name in our system to add specific details on the author's contribution. More information is available in our guide to authors.

16) As part of the EMBO Publications transparent editorial process initiative (see our Editorial at <http://embomolmed.embopress.org/content/2/9/329>), EMBO Molecular Medicine will publish online a Review Process File (RPF) to accompany accepted manuscripts.

In the event of acceptance, this file will be published in conjunction with your paper and will include the anonymous referee reports, your point-by-point response and all pertinent correspondence relating to the manuscript. Let us know whether you agree with the publication of the RPF and as here, if you want to remove or not any figures from it prior to publication. Please note that the Authors checklist will be published at the end of the RPF.

I look forward to receiving your revised manuscript.

With kind regards,

Lise Roth

***** Reviewer's comments *****

Referee #1 (Comments on Novelty/Model System for Author):

In this study, the authors analyzed CBX7 and klf2 (related downstream genes) with CCM2 in Zebrafish model, but some data were from CCM1. For data consistency, I would like to see data and conclusion in both CCM1 and CCM2 models if possible, otherwise, the authors could discuss the differences in discussion section.

Referee #1 (Remarks for Author):

In this manuscript, Pham et al. investigated the novel role of the epigenetic modifier Cbx7a activated by Klf2 in Zebrafish CCM model, with a focus on its cardiovascular phenotype. Further the authors explored its downstream target genes which contribute CCM phenotype. Some experiments were carefully conducted, whereas some critical data were confusing and incomplete. The conclusions were not satisfactory, and I do not think the current manuscript is suitable for publication in the Journal of EMBO Molecular Medicine.

- 1, This study employed CCM2 model and conclusions were made with CCM2 in most of data while some were done with Krit1/CCM1 (e.g. Fig.1F, Fig.2Q-T, Fig.S3A-E). This raises questions about whether CCM2/Krit1 share same signaling pathways. If so, the authors should perform both CCM2 and Krit1/CCM1 model together in all data sets. Or some data were not detected in both CCM2 and Krit1 model? For example, in Fig.S3A-E, it is unclear what happens with wnt9b level in CCM2m201 with Klf2a/b MO. In addition, the expression level of wnt9b in Fig.S3B and Fig.S3D showed opposite results compared to the quantification data in Fig.2C which wnt9b was lower expressed in CCM2 than in Krit1.
- 2, CBX7 was elevated in CCM1/CCM2 of Zebrafish heart, as well as in mouse BMCEs and Human iPS-derived ECs as shown in Fig.1B, did the author check CBX7 expression in human lesion as shown in Fig.2U? In this same data set (Fig.2U), in addition to wnt9b, it would be valuable to investigate changes in CBX7/Klf2 downstream genes such as tek/angpt1 in human CCM lesion.
- 3, The authors concluded that Klf2 positively regulates Cbx7, while blood flow negatively controls Cbx7 (Fig.1M-T; Fig.S2) based on Klf2 overexpression in endothelial cells and tnt2a MO experiments. However, based on Fig.S2 A and C, it looks like blood flow may only plays a similar role as Klf2 in cardiovascular defects. Is Klf2 important in CCM2 model? Are Cbx7, wnt9b, tek and angpt1 changed with blood flow condition without endothelial cell specific klf2 overexpression?
- 4, In the material and methods section, the author should provide much more detailed and thorough information on the quantitative analysis conducted and presented in the graphs. This is important for the purpose of data reproducibility and research integrity. For example, in Fig.1L and Fig.S1C, it is unclear how endocardial cells were counted and normalized whether it was by per heart, volume, or area. Additionally, it was unclear how the Y axis of "mRNA level" in all other graphs was normalized. It is important to assess whether there are significant differences in Fig.S1B, Fig.2T and Fig.3B-D' as the authors drew some conclusions based on these data (e.g. line 191). For example, the statement "double mutants normalized heart morphology and function (Fig.1K; Fig.S1), which restored blood flow in a subset of these double mutants (Fig.S1B)". could be supported by statistical analysis such as Chi-square test.
- 5, In lines 289-293, the authors claimed that angpt1 and tek were normalized to wild type level upon knock down of Cbx7a (Fig.2K, L) in endothelial cell specific Klf2 transgenic embryos. However, it appears that this is not case for angpt1, as shown in Fig.2K (graph in red color), indicating that angpt1 is not one of klf2 target genes regulated by Cbx7?
- 6, Fig.S1D-L and lines 198-200: it would be much more convincing if there were quantification data to support the conclusion that "other CCM-related defects in different vascular beds...were restored upon a loss of Cbx7a."
- 7, In this manuscript, I did not find any data related to in vivo mouse models, please remove "In vivo experiments and animal model sections" from "material and methods section". Additionally, the karyotyping results and methods for Fig.S4 F-H were not described in manuscript.
- 8, Based on current data, it is still unsatisfactory with the signaling pathway model in Fig.3R. it would be beneficial to provide more details about the relationship among blood flow, Klf2, Cbx7, wnt9b, Tek and angpt1?

other minor comments:

- 1, Fig.S1C: there is no significant difference between wild type and cbx7a mutant?
- 2, line 279, Fig.3U should be "Fig.2U"

3, line289. Fig.2K, L should be Fig.2J,L

4, Line1342, "one-way OVONA" should be "one-way ANOVA"

Referee #2 (Comments on Novelty/Model System for Author):

Analyses performed using morphosinos should be replaced by using mutants.

Referee #2 (Remarks for Author):

In this manuscript, Pham et al identify the epigenetic modifier CBX7 as an important contributor to the pathophysiology of CMM genes during ZF heart development. Overall, the study is well performed, and well-written; yet it misses several controls and validations.

Main relevant points which should be discussed/addressed are:

The *ccm2* and *Krit1* mutant are constitutive mutants; how do authors differentiate between a phenotype in the heart per se and a putative phenotype in ECs?

The data that *cbx7a* is a pathological downstream effector in *ccm2* and *Krit1* in the heart are convincing. Yet, most of the experiments have been done in the heart as a model. While this is an interesting system to study *ccm2* and *Krit1*, it does not represent the human CCM diseases. So, authors should be very careful with the statement they make across the manuscript. For instance, the models presented here are not a ZF model of CCM (line 162). Is CBX7A relevant for the human CCM disease? Or is it simply important for heart development?

Another major point is that authors use across the manuscript the *Tg(fli1a:Gal4FF)ubs3;Tg(UAS:klf2a)ig1* to claim that the molecular mechanism and phenotypes imposed by CCM genes are *klf2* mediated. For making this point, authors should study better the *ccm2/klf2* compound mutants (they also have them; Fig. 1P).

Quantification of *cbx7a* mRNA expression levels in *Tg(fli1a:Gal4FF)ubs3;Tg(UAS:klf2a)ig1* with blood flow defects vs. normal should be shown. Expression levels of KLF2 should be also shown in Fig 1B. It would be nice if authors would show that in HUVECs or iPCS cells that Klf2 regulates CBX7A expression.

The morpholino approach has been largely disputed. Hence, for some of the claims of the manuscript should use mutants instead of morpholinos (i.e. *Klf2a* and *klf2b* and *wnt9b* mutant). In fact, this lab has already published some work using the *wnt9b* mutant, so why did they not use it here? This is especially relevant given that the *wnt9DKO* has a phenotype while the *wnt9bMO* does not.

How do authors explain that *wnt9b* and *9a* ligands are much higher upregulated in *Krit1ty219c* compared to *ccm2m201* while *cbx7a* mRNA levels show the opposite pattern? Fig 2C vs. Fig. 1F. Whole-mount *wnt9b* in situ experiments on the *ccm2m201* and *Krit1ty219c* should be shown in Fig. 2C

Again, why do authors show *ccm2* and *Krit1* MO instead of mutants in Fig. 2Q-S? For consistency across the manuscript, authors should show primary focus on *ccm2*...

In line 279 should say Fig. 2U, and not Fig. 3U

In line 309, authors state: "This demonstrated that the Tie1/Tek signaling pathway is downstream of *Klf2a* in zebrafish models of CCM". This is incorrect as they do not show this" They simply show that *Krit1MO* in a Tie1/Tie2 background restores the phenotype or that BAY826 restores the phenotype in *ccm2m201* mutants.

Data showing siRNA depletion of CCM2 in HUVECs are missing. Similarly, data showing that ECs derived from iPCS are KO for CCM2 should be shown. Is the junctional phenotype shown in siCCM2 HUVECs also found in vivo? Authors should show the VE-Cadherin staining split from the rest. It is hard to appreciate as it is.

Do the transcripts identified in the RNAseq correlate with the CUT&RUN-seq analysis of H3K27me3 marks?

Referee #3 (Comments on Novelty/Model System for Author):

The manuscript by Pham et al describes the role of CBX7, a component of the Polycomb Repressive Complex1, during CCM

signaling in zebrafish embryos and human endothelial cells. The authors demonstrated that Cbx7 homolog expression is upregulated in CCM-deficient zebrafish embryos and human cells, and that mutation or inhibition of Cbx7 function reverses CCM phenotypes. They utilize transcriptomic approaches to demonstrate that Cbx7 plays a role in activation of Klf2 targeting genes including Wnt9 and others. Altogether, this study reveals a mechanistic role for CBX7 during CCM pathogenesis. Overall these results are novel and highly significant, as they contribute to mechanistic understanding of CCM pathogenesis. Experiments are well performed, and results are clear and strongly support conclusions.

Referee #3 (Remarks for Author):

Overall these results are novel and highly significant, as they contribute to mechanistic understanding of CCM pathogenesis. Experiments are well performed, and results are clear and strongly support conclusions. It is suggested that the authors address the following comments.

1. Cbx7 upregulation in the heart of *ccm2* and *krit1* mutants is clear and very impressive. The authors claim that it is expressed in the endocardium, although this is not immediately clear from the provided images. It would be helpful to confirm the endocardial expression using double labeling with endothelial marker, such as *kdr1:GFP*.
2. Similar *cbx7* upregulation in the head and tail tissue was observed in *ccm2* or *krit1* mutants (Fig. 1F). Is this upregulation specific to vascular endothelial cells? In situ hybridization combined with endothelial marker or qPCR from FACS-sorted endothelial cells could be used to answer this question.
3. Rescue of *ccm2* phenotype in double *ccm2; cbx7a* mutants nicely demonstrates that *cbx7a* upregulation is required for cardiac defects in *ccm2* mutants. Is *cbx7a* upregulation sufficient to cause CCM-like phenotype? This could be done by inducing *cbx7a* expression using endothelial specific promoter. While I would consider this experiment somewhat optional, this could provided an important insight into *cbx7a* function and CCM pathogenesis.
4. Authors state that endocardial cell numbers were not significantly reduced in *cbx7a* mutants (Fig. S1C). Yet it appears from the graph that there could be some reduction in endocardial cell numbers, which perhaps was not statistically significant due to limited n. Could the authors analyze more embryos to see if this reduction could be significant? Also, please provide p value for this experiment, even if it is not significant.
5. Labeling in Fig. 2A is confusing. There are sets of genes going up (in *ccm2* mutants) and down in (*ccm2; cbx7a* mutants), and the other way around. Is this relative to wt embryos? The title says "...genes misregulated in *ccm2* mutants compared to *ccm2; cbx7a*"
6. CUT&RUN experiment discussed in Fig. 3P,Q is not discussed in any detail. I could not find a description of this experiment in Methods section either. Please provide more detailed description of this experiment and relevant analysis.
7. Fig. 2U shows upregulation of WNT9B and WNT9A in human CCM lesions and iPSC-derived ECs. It is not clear what the control is here. What is the expression compared to?

Minor points.

1. Please check the reference to Fig. 3U on page 11, line 279; perhaps it should be Fig. 2U.
2. Authors describe in the discussion that zebrafish *cbx7a* mutants were phenotypically normal (Fig. S5). This data should be presented much earlier in the results section, when *cbx7a* mutant generation is first described.
3. Please check the reference to Fig. S5 in the Methods section, page 30, line 784; Fig. S5 does not show karyotyping.

We would like to thank the reviewers for their efforts to review our manuscript. Their comments and suggestions have improved its quality. We hope that with these changes and improvements, our reviewers will find this manuscript suitable for publication with EMBO Molecular Medicine.

Referee #1 (Comments on Novelty/Model System for Author):

In this study, the authors analyzed CBX7 and *klf2* (related downstream genes) with CCM2 in Zebrafish model, but some data were from CCM1. For data consistency, I would like to see data and conclusion in both CCM1 and CCM2 models if possible, otherwise, the authors could discuss the differences in discussion section.

In our study, we had initially laid focus on the zebrafish CCM2 model, because the CCM1, CCM2, and CCM3 proteins are part of the same protein complex and mutations in any of these genes result in similar disruptions of endothelial cell function. This leads to the formation of CCMs with similar structural and functional characteristics. In patients, this manifests as similar clinical symptoms regardless of which of the three CCM genes is mutated. These symptoms include seizures, headaches, and neurological deficits caused by the abnormal blood vessels.

Within our revised manuscript, we have now expanded our analyses and provide new data from a number of CCM mutants:

-In Fig. 1G-K, we show that CBX7 protein is found in lesion material from CCM1 and CCM2 patients.

-In Figure EV2N-Q, we show that the knockdown of *cbx7a* suppresses *krit1* mutant phenotypes in zebrafish.

-In Fig. 1F, we show that *cbx7a* mRNA is upregulated in *ccm2* and *krit1* zebrafish mutants. Also, in Figure EV4D-F, we show that *cbx7a* expression is upregulated in *ccm2* mutant hearts.

-In Figure EV4G-N, we show that the loss of *wnt9b* alleviates the loss of *krit1* and *ccm2* mutant phenotypes in zebrafish.

-Finally, we also perform pharmacological suppression studies in *ccm2* zebrafish mutants (Fig. 4B-D) and in endothelial-specific CCM3 knockout mice (Fig. 4E-J). Of note, CCM3 represents the most aggressive form of CCM in patients.

Together, this new data strongly supports our main finding of epigenetic regulation by the Polycomb Repressive Complex 1 in CCM.

Referee #1 (Remarks for Author):

In this manuscript, Pham et al. investigated the novel role of the epigenetic modifier Cbx7a activated by Klf2 in Zebrafish CCM model, with a focus on its cardiovascular phenotype. Further the authors explored its downstream target genes which contribute CCM phenotype. Some experiments were carefully conducted, whereas some critical data were confusing and incomplete. The conclusions were not satisfactory, and I do not think the current manuscript is suitable for publication in the Journal of EMBO Molecular Medicine.

1, This study employed CCM2 model and conclusions were made with CCM2 in most of data while some were done with Krit1/CCM1 (e.g.Fig.1F, Fig.2Q-T, Fig.S3A-E). This raises questions about whether CCM2/Krit1 share same signaling pathways. If so, the authors should perform both CCM2 and Krit1/CCM1 model together in all data sets. Or some data were not detected in both CCM2 and Krit1 model? For example, in Fig.S3A-E, it is unclear what happens with wnt9b level in CCM2m201 with Klf2a/b MO.

KRIT1/CCM1, CCM2, and CCM3 proteins are part of the same complex and pathway, mutations in any of these genes result in similar disruptions to endothelial cell function. It is well established that KRIT1 and CCM2 signal via the KLF2 pathway.

In addition, within the revised manuscript we have included additional data on relevant targets or function of Cbx7a for both mutants/morphants of Krit1 and Ccm2, respectively.

-Fig. 1F: Expression levels of *cbx7a* mRNA in different tissues of *krit1^{ly219c}* mutants are now provided in addition to levels in *ccm2^{m201}* mutants.

-Fig. EV4C,F: We have now included representative images of whole mount *in situ* hybridisation of *wnt9b* transcripts in *ccm2*; *klf2a/klf2b* double morphant embryos. This reveals that *cbx7a* mRNA levels are reduced upon loss of Klf2a/b in this zebrafish *ccm2* model.

In addition, the expression level of wnt9b in Fig.S3B and Fig.S3D showed opposite results compared to the quantification data in Fig.2C which wnt9b was lower expressed in CCM2 than in Krit1.

We suggest to rely on qPCR quantifications rather than Whole mount *in situ* hybridisation images for quantifying gene expression levels. The latter one, we generally use to detect patterns of expression rather than quantifiable levels of expression. Expression levels of wnt9b are now provided in Fig. 3C of our revised manuscript.

2, CBX7 was elevated in CCM1/CCM2 of Zebrafish heart, as well as in mouse BMCEs and Human iPS-derived ECs as shown in Fig.1B, did the author check CBX7 expression in human lesion as shown in Fig.2U?

In the revised manuscript we have we have added immunohistological stainings of CBX7 in lesion material from familial CCM patients. This revealed strong expression of CBX7 in lesional ECs and surrounding neural tissues of all patients with familial CCM1 and CCM2 (Fig. 1G-J).

In this same data set (Fig.2U), in addition to wnt9b, it would be valuable to investigate changes in CBX7/Klf2 downstream genes such as *tek/angpt1* in human CCM lesion.

We did not detect upregulation of TEK or ANGPT1 in the transcriptomic data set show in Fig. 2U. However, we would like to point out that surgically resected lesion material can be quite heterogeneous and contains a significant amount of other brain tissues apart from the endothelial cells. Furthermore, endothelial cells lining such lesions are a compound of CCM-null endothelial cells and many wild-type cells that have been attracted to contribute to the cavernoma growth (Malinverno et al., 2019). Such wild-type cells within lesions may impact the outcome of qPCR analyses and sequencing data.

3, The authors concluded that Klf2 positively regulates Cbx7, while blood flow negatively controls Cbx7 (Fig.1M-T; Fig.S2) based on Klf2 overexpression in endothelial cells and *tnnt2a* MO

experiments. However, based on Fig.S2 A and C, it looks like blood flow may only plays a similar role as Klf2 in cardiovascular defects. Is Klf2 important in CCM2 model?

In the context of CCM, KLF2 and its orthologue KLF4 are key regulators that become overexpressed in CCM-deficient endothelial cells. We would like to point out that we summarize the state of the art in our introduction:

“The loss of any of these proteins enhances signaling via MAP3K3/ERK5 and Krüppel-like transcription factors KLF2/4. This is a key regulatory pathway of biomechanical responses within endothelial cells. Functional studies have shown that CCM-associated cardiovascular defects can be suppressed when CCM-deficient mice undergo a loss of Klf2/4 or zebrafish *ccm* mutants lack Klf2a/b. Klf2 can trigger an endothelial-to-mesenchymal transition (endMT), which is thought to contribute to the formation of CCM lesions because endothelial cells acquire mesenchymal properties. These discoveries have established KLF2 signaling as a key player in CCM, but its downstream targets in the etiopathology of the disease are largely unknown.”

Here, we show that, in the presence of blood flow, Klf2a activates different levels of Cbx7a in the presence compared to the absence of blood flow (quantified in Fig. 2S and shown in Fig. 2.Q,R).

Are Cbx7, wnt9b, tek and angpt1 changed with blood flow condition without endothelial cell specific klf2 overexpression?

We have currently no evidence for an activation of these genes by blood flow in a Klf2-independent manner. This would be a challenging study, which would be beyond the scope of this manuscript.

4, In the material and methods section, the author should provide much more detailed and thorough information on the quantitative analysis conducted and presented in the graphs. This is important for the purpose of data reproducibility and research integrity. For example, in Fig.1L and FigS1C, it is unclear how endocardial cells were counted and normalized whether it was by per heart, volume, or area. Additionally, it was unclear how the Y axis of "mRNA level" in all other graphs was normalized. It is important to assess whether there are significant differences in Fig.S1B, Fig.2T and Fig.3B-D' as the authors drew some conclusions based on these data (e.g. line 191). For example, the statement "double mutants normalized heart morphology and function (Fig.1K; Fig.S1), which restored blood flow in a subset of these double mutants (Fig.S1B)". could be supported by statistical analysis such as Chi-square test.

In Fig. 1L and Fig S1C (now Fig. 2F and Fig. S1D), the number of endocardial cells were quantified by using Spot function in Imaris, and each data point represented in the graphs is from an individual heart/embryo.

For mRNA quantifications by qRT-PCR, we have mentioned in Material and Method section that the expression level of gene of interest was normalized with *eif1b* as internal control, following the $2^{-\Delta\Delta C1}$ method.

Regarding the comment on "restore subset of blood flow", in Fig. EV2A, each column represents a genetic group. It is important to keep in mind that in zebrafish *ccm* mutants, the heart is dilated and fails to establish blood flow. Hence, the value for "normalized heart-with Blood Flow" in *ccm2,cbx7a+/+* is expected to be 0, which is not suitable to be analyzed by the Chi-square test.

5, In lines 289-293, the authors claimed that *angpt1* and *tek* were normalized to wild type level upon knock down of *Cbx7a* (Fig.2K, L) in endothelial cell specific *Klf2* transgenic embryos. However, it appears that this is not case for *angpt1*, as shown in Fig.2K (graph in red color), indicating that *angpt1* is not one of *klf2* target genes regulated by *Cbx7*?

Within the revised manuscript, we have now summarized these findings as follows:

“The upregulation of *tek* was normalized to wild-type levels upon knockdown of *Cbx7a* (Fig. 3Q), while levels of *angpt1* were not significantly reduced (Fig. 3P).”

While the data show a tendency of *angpt1* levels to decrease upon knockdown of *Cbx7a*, the data is not significant yet.

6, Fig.S1D-L and lines 198-200: it would be much more convincing if there were quantification data to support the conclusion that "other CCM-related defects in different vascular beds...were restored upon a loss of *Cbx7a*."

We have added quantifications of these phenotypes and their rescue that are now shown in Figure EV2D-M.

7, In this manuscript, I did not find any data related to in vivo mouse models, please remove "In vivo experiments and animal model sections "from "material and methods section".

This has been corrected. We are now providing additional data on a curative mouse model of CCM3 using a CBX7-specific inhibitor (Fig. 4E-J).

Additionally, the karyotyping results and methods for Fig.S4 F-H were not described in manuscript.

We now report in the material and methods section:

“cells were prepared for karyotyping accordingly to the protocol provided by Cell Guidance Systems and shipped to them for analysis. Karyotyping was evaluated by g-banding (Fig. S4).”

Results are reported in the Figure legend in Fig. S2F,G: “Karyotyping results of WT and CCM2 KO iECs indicate that there are no chromosomal anomalies associated with the knockout procedure and that healthy clones were generated and utilized.”

8, Based on current data, it is still unsatisfactory with the signaling pathway model in Fig.3R. it would be beneficial to provide more details about the relationship among blood flow, *Klf2*, *Cbx7*, *wnt9b*, *Tek* and *angpt1*?

We have added the factors WNT9B, TEK, and ANGPT1 to the model in Fig. 4K to better emphasize the role of these factors in our proposed model.

other minor comments:

1, Fig.S1C: there is no significant difference between wild type and *cbx7a* mutant?

Within the revised manuscript, we added the p value, showing no significance.

2, line279, Fig.3U should be "Fig.2U

We have corrected this error in the revised manuscript.

3, line289. Fig.2K, L should be Fig.2J,L

We have corrected this error in the revised manuscript.

4, Line1342, "one-way OVONA" should be "one-way ANOVA"

We have corrected this error in the revised manuscript.

Referee #2 (Comments on Novelty/Model System for Author):

Analyses performed using morphosinos should be replace by using mutants.

Referee #2 (Remarks for Author):

In this manuscript, Pham et al identify the epigenetic modifier CBX7 as an important contributor to the pathophysiology of CMM genes during ZF heart development. Overall, the study is well performed, and well-written; yet it misses several controls and validations.

Main relevant points which should be discussed/addressed are:

The ccm2 and Krit1 mutant are constitutive mutants; how do authors differentiate between a phenotype in the heart per se and a putative phenotype in ECs?

We previously demonstrated that the CCM complex has a tissue-specific effect within the endothelium/endocardium. Restoring the activity of Krit1 specifically within endocardial cells rescued the heart phenotypes and many vascular defects present in the endothelium of these constitutive mutants (Rödel et al. 2019 Circ Res). These findings are discussed in the introduction section:

“Zebrafish completely lacking Krit1 protein exhibited severe defects in the heart, preventing blood flow^{15,19,20}. In this physiological condition without blood flow, even major blood vessels developed CCM phenotypes^{15,19–21}. Strikingly, expressing Krit1 specifically in the heart restored blood flow and suppressed the pathological phenotypes in these major blood vessels²⁰. This suggests that high levels of blood flow exert protective effects, in contrast to regions of low or no blood flow where CCM lesions occur.”

The endocardium is a specialized endothelium that recapitulates all properties of endothelial cells such as barrier function, selective permeability, and the secretion of bioactive substances, while also being uniquely adapted to support cardiac-specific functions like modulating myocardial contractility and conducting electrical impulses. Major hallmarks of CCM key functional players

such as MEKK3-KLF2/4 pathway, Integrin and ICAP1, and VEGF signalling have been identified in zebrafish CCM models.

Our findings now demonstrate that CBX7 is upregulated specifically with the endothelium/endothelium due to the activity of KLF2 in different CCM models and that silencing of *Cbx7a*/CBX7 suppresses CCM phenotypes in zebrafish endothelial tissues. We demonstrate the tissue-specificity of these effects also in an endothelial-specific knockout model for CCM3.

The data that *cbx7a* is a pathological downstream effector in *ccm2* and *Krit* in the heart are convincing. Yet, most of the experiments have been done in the heart as a model. While this is an interesting system to study *ccm2* and *Krit*, it does not represent the human CCM diseases. So, authors should be very careful with the statement they make across the manuscript. For instance, the models presented here are not a ZF model of CCM (line 162). Is CBX7A relevant for the human CCM disease? Or it is simply important for heart development?

Having found upregulation of CBX7 also in CCM-deficient HUVECs and human iPSC-derived ECs encouraged us investigate the relevance of this factor both in the context of zebrafish heart development as well as CCM pathology. CCM protein play a crucial role during heart and cardiac valve development and findings in this system have often been found to be also relevant in the disease condition (Renz and Otten, et al., 2015; Zhou and Rawnsley, et al., 2015; Abou-Fadel and Zhang, 2020; Philips et al., 2022). For instance, in Renz et al. 2015, we identified the role of KLF2 as functionally relevant in the expression of CCM phenotypes in zebrafish and human endothelial cells depleted of CCM proteins.

In the revised manuscript, we have now also included immunohistological stainings of CBX7 in lesion material from familial CCM patients. This reveals strong expression of CBX7 in lesional ECs and surrounding neural tissues of patients with familial CCM1 and CCM2 (Fig. 1G-J).

We have also included the curative effects of using a CBX7-specific inhibitor in an endothelial-specific CCM3 knockout mouse model, showing that silencing of CBX7 suppresses lesion size and number. Taken together, these are encouraging data suggesting that CBX7 has relevance for the human CCM disease.

Another major point is that authors use across the manuscript the *Tg(fli1a:Gal4FF)ubs3;Tg(UAS:klf2a)ig1* to claim that the molecular mechanism and phenotypes imposed by CCM genes are *klf2* mediated. For making this point, authors should study better the *ccm2/klf2* compound mutants (they also have them; Fig. 1P).

We would like to thank the reviewer for this insightful comment. We have used *Klf2* gain-of-function experiments as a means to identify KLF2 target genes affected by an activity of *Cbx7a*:

“Next, we aimed at elucidating whether the PRC1 component *Cbx7a* had an effect on *Klf2* target gene activation. In pursuit of this objective, we devised a transcriptomic approach to identify *Cbx7a*-dependent target genes of *Klf2a* that (1) exhibit an upregulation in *ccm* mutants, (2) show at least partial normalization in *ccm2;cbx7a* double mutants, and (3) are inducible by *Klf2*.”

However, we have also included additional data sets using *Klf2* loss of function conditions. For instance, we now show the role of *Klf2a/b* in the context of *wnt9b* expression in *krit1* and *ccm2* mutants in Fig. EV4C,F.

Quantification of *cbx7a* mRNA expression levels in Tg(*fli1a*:Gal4FF)*ubs3*;Tg(UAS:*klf2a*)*ig1* with blood flow defects vs. normal should be shown.

This is now shown in revised Fig. 2S (and WISH thereof in Fig. 2Q,R)

Expression levels of KLF2 should be also shown in Fig 1B.

Levels of *klf2a* mRNA in *ccm2* mutants have been published in Renz et al., 2015 and this data is referenced in the figure legend.

It would be nice if authors would show that in HUVECs or iPCS cells that *Klf2* regulates CBX7A expression.

Unfortunately, we have not been able to perform such an experiment in human endothelial cells.

The morpholino approach has been largely disputed. Hence, for some of the claims of the manuscript should use mutants instead of morpholinos (i.e. *Klf2a* and *klf2b* and *wnt9b* mutant). In fact, this lab has already published some work using the *wnt9b* mutant, so why did they not use it here? This is especially relevant given that the *wnt9DKO* has a phenotype while the *wnt9bMO* does not.

This is an important point and clearly needs to be addressed, as it is true that morpholinos have to be used according to clear recommendations (Stainier et al., 2017; “Guidelines for morpholino use in zebrafish”). Here, and in Paolini et al 2023 (<https://doi.org/10.1242/dev.201707>), the *wnt9b* morpholino was verified according to these recommendations. The *wnt9a* and *wnt9b* morpholinos were carefully verified to ensure the reproduction of the double knock-out mutant phenotype (Paolini et al., 2023: “To further verify these findings, we used an antisense oligonucleotide morpholino (MO) knockdown approach against *Wnt9a/b*. This produced a cardiac phenotype identical to that seen in *wnt9DKO* mutants at 48 hpf (Fig. S2).”). Paolini et al. also show that *wnt9b^{sa20083}* and *wnt9a^{sd49}* single mutants lack cardiac defects (Fig. S1). Similarly, single knock-down using either *wnt9a* or *wnt9b* morpholino did not cause a phenotype.

To further substantiate our data, we generated *ccm2^{m201}*; *wnt9b^{sa20083}* double mutant embryos and found that this caused a restoration of cardiac ballooning and loss of ALCAM expression in endocardial cells of the AVC in *ccm2^{m201}* mutants. These data are now present in the revised manuscript in (Fig. EV4M).

How do authors explain that *wnt9b* and 9a ligands are much higher upregulated in *Krit1ty219c* compared to *ccm2m201* while *cbx7a* mRNA levels show the opposite pattern? Fig 2C vs. Fig. 1F.

At this point, we can only speculate as to why these diametrical expression profiles of *cbx7a* and *wnt9b* occur in the *krit1* and *ccm2* mutant backgrounds. For instance, differential levels of *Wnt9b* expression may dependent not only on the levels of *Klf2a*, but also activity of the PRC1 and on the number of *Klf2a* transcription factor binding sites in the promoter regions and other regulatory sites of *wnt9b*. Currently, we have no satisfactory explanation to the differences in transcriptional activation and use a conservative statement to describe *Cbx7a* and *Wnt9b* expression levels, reporting them as increased in *ccm2^{m201}* and *krit1^{ty219c}* mutants as compared to wild-type.

Whole-mount *wnt9b* in situ experiments on the *ccm2m201* and *Krit1ty219c* should be shown in Fig. 2C

These experiments are now shown in Fig. EV4A-F.

Again, why do authors show *ccm2* and Krit1 MO instead of mutants in Fig. 2Q-S? For consistency across the manuscript, authors should show primary focus on *ccm2*...

Unfortunately, there is no working *ccm2* antisense morpholino oligo available. We were therefore not able to repeat the experiment shown in Fig. 3W-Y in the *tek^{hu1667}; tie1^{bsn208}* mutant background. However, as we have mentioned above, KRIT1/CCM1, CCM2, and CCM3 proteins are part of the same complex and pathway, mutations in any of these genes result in similar disruptions to endothelial cell function.

In line 279 should say Fig. 2U, and not Fig. 3U

We have corrected this error.

In line 309, authors state: "This demonstrated that the Tie1/Tek signaling pathway is downstream of Klf2a in zebrafish models of CCM". This is incorrect as they do not show this" They simply show that Krit1MO in a Tie1/Tie2 background restores the phenotype or that BAY826 restores de phenotype in *ccm2m201* mutants.

We would like to thank the reviewer for pointing out this over-interpretation of data. We have changed the revised manuscript as follows:

"The cardiac ballooning phenotype in *krit1* morphants decreased significantly in the presence of mutant alleles, particularly of *tie1*, and *krit1* phenotypes were most strongly suppressed in *tek^{hu1667}; tie1^{bsn208}* double mutants (Fig. 3W-Y). This demonstrated that Tie1/Tek signaling is involved in cardiovascular phenotypes in zebrafish models of CCM."

Data showing siRNA depletion of CCM2 in HUVECs are missing.

The siRNA depletion of CCM2 is highly efficient as previously published in Jilkova et al., 2014.

Similarly, data showing that ECs derived from iPCS are KO for CCM2 should be shown.

This evidence is shown in Fig. S2C by demonstrating that a large deletion resulting in a frame shift and premature stop codon is induced in these cells.

Is the junctional phenotype shown in siCCM2 HUVECs also found in vivo?

Other studies published the effects of loss of CCM proteins on EC junctional integrity (i.e. reviewed in Lampugnani et al., 2017 PMID: 28851747).

Authors should show the VE-Cadherin staining split from the rest. It is hard to appreciate as it is.

Within revised Fig. 2H-K, we have now separated Ve-Cadherin staining from F-actin labelling.

Do the transcripts identified in the RNAseq correlate with the CUT&RUN-seq analysis of H3K27me3 marks?

Within revised Fig. EV5, we show GO term clustering of RNAseq and CUT&RUNseq data sets (for both suppressive H3K27me3 marks and activating H3K4me3 marks). This comparison reveals a great overlap in regulated processes and indicates the role of epigenetic modification in CCM hallmark gene expression.

We have included a better explanation of the GO term analysis in the revised version of the manuscript with respect to the RNAseq data and CUT&RUNseq data sets.

Referee #3 (Comments on Novelty/Model System for Author):

The manuscript by Pham et al describes the role of CBX7, a component of the Polycomb Repressive Complex1, during CCM signaling in zebrafish embryos and human endothelial cells. The authors demonstrated that Cbx7 homolog expression is upregulated in CCM-deficient zebrafish embryos and human cells, and that mutation or inhibition of Cbx7 function reverses CCM phenotypes. They utilize transcriptomic approaches to demonstrate that Cbx7 plays a role in activation of Klf2 targeting genes including Wnt9 and others. Altogether, this study reveals a mechanistic role for CBX7 during CCM pathogenesis. Overall these results are novel and highly significant, as they contribute to mechanistic understanding of CCM pathogenesis. Experiments are well performed, and results are clear and strongly support conclusions.

We would like to thank the reviewer for recognising the novelty and mechanistic relevance of our findings.

Referee #3 (Remarks for Author):

Overall these results are novel and highly significant, as they contribute to mechanistic understanding of CCM pathogenesis. Experiments are well performed, and results are clear and strongly support conclusions. It is suggested that the authors address the following comments.

1. Cbx7 upregulation in the heart of *ccm2* and *krit1* mutants is clear and very impressive. The authors claim that it is expressed in the endocardium, although this is not immediately clear from the provided images. It would be helpful to confirm the endocardial expression using double labeling with endothelial marker, such as *kdrl*:GFP.

The reviewer is correct in his observation that tissue-specific localisation on *cbx7a* mRNA was not clearly identifiable. To confirm the expression of Cbx7a in the endocardium, we performed fluorescent *in situ* hybridisations, which revealed a clear localisation in endocardial cells only (Fig. EV1). Furthermore, RNASeq analysis of human iPSC-derived endothelial cells revealed an upregulation of CBX7, suggesting that CBX7 is expressed in in this tissue (Fig. 1B). Finally, we now provide additional expression data for human CBX7 in lesion material from familial CCM patients. These immunohistological stainings show a strong localisation CBX7 in endothelial cells, as well as surrounding tissues (Fig. 1G-K).

2. Similar *cbx7* upregulation in the head and tail tissue was observed in *ccm2* or *krit1* mutants (Fig. 1F). Is this upregulation specific to vascular endothelial cells? In situ hybridization combined with endothelial marker or qPCR from FACS-sorted endothelial cells could be used to answer this question.

As pointed out above, RNASeq analysis of human iPSC-derived endothelial cells revealed an upregulation of CBX7 (Fig. 1B). Finally, we now provide additional expression data for human CBX7 in lesion material from familial CCM patients (Fig. 1G-J). These immunohistological stainings show a strong localisation CBX7 in endothelial cells, as well as other adjacent tissues.

3. Rescue of *ccm2* phenotype in double *ccm2*; *cbx7a* mutants nicely demonstrates that *cbx7a* upregulation is required for cardiac defects in *ccm2* mutants. Is *cbx7a* upregulation sufficient to cause CCM-like phenotype? This could be done by inducing *cbx7a* expression using endothelial specific promoter. While I would consider this experiment somewhat optional, this could provide an important insight into *cbx7a* function and CCM pathogenesis.

We would like to thank the reviewer for this excellent suggestion. However, we have not been able to perform such a *Cbx7a* gain of function experiment. However, indirectly, this experiment was done when we demonstrated that *Tg(fli1a:GAL4FF)^{ubs3}*; *Tg(UAS:klf2a)^{ig1}* also induces *Cbx7a* (Fig. 2R), which causes a cardiac dilation in the atrium.

4. Authors state that endocardial cell numbers were not significantly reduced in *cbx7a* mutants (Fig. S1C). Yet it appears from the graph that there could be some reduction in endocardial cell numbers, which perhaps was not statistically significant due to limited n. Could the authors analyze more embryos to see if this reduction could be significant? Also, please provide p value for this experiment, even if it is not significant.

To more robustly assess statistical significance, we increased the number of embryos analyzed in Fig. S1D and found that, indeed, there is a statistically significant slight decrease in endocardial cell numbers in *cbx7a* mutants. However, we did not observe any gross morphological differences of the cardiovascular system.

5. Labeling in Fig. 2A is confusing. There are sets of genes going up (in *ccm2* mutants) and down in (*ccm2*; *cbx7a* mutants), and the other way around. Is this relative to wt embryos? The title says "...genes misregulated in *ccm2* mutants compared to *ccm2*; *cbx7a*"

We agree with the reviewer that this figure and data presentation needed to be simplified. To do so, we have changed the data presentation and the text as follows and hope that the logic of this experiment is easier to follow:

"When comparing the cardiac transcriptome data sets, we found 1330 genes upregulated in *ccm2^{m201}* compared to wild-type. Of these, 429 genes were overlapping with genes that were also upregulated in *ccm2^{m201}* mutants compared to *ccm2^{m201}*; *cbx7a^{pb62}* double mutants. Gene ontology analyses of this set of 429 genes indicated an enrichment of terms related to angiogenesis, cell migration, endothelial proliferation, endoMT, Wnt signaling, and other processes, some of which had previously been associated with the CCM pathology (Fig. 3A)^{1,15,17,20,47-49}. Conversely, 1413 genes were downregulated in *ccm2^{m201}* compared to wild-type, of which 513 genes were also downregulated in *ccm2^{m201}* compared to *ccm2^{m201}*; *cbx7a^{pb62}* double mutants. This set of 513 genes was enriched for gene ontology terms related to cell adhesion, ECM organization, and EMT processes (Fig. 3A). Among the genes associated with the biological processes angiogenesis, canonical Wnt signaling and EndoMT were several that had previously been implicated in KLF2/4 signaling (Fig. 3B)^{12,50}."

6. CUT&RUN experiment discussed in Fig. 3P,Q is not discussed in any detail. I could not find a description of this experiment in Methods section either. Please provide more detailed description of this experiment and relevant analysis.

Within revised Fig. EV5, we show GO term clustering of RNAseq and CUT&RUNseq data sets (for both suppressive H3K27me3 marks and activating H3K4me3 marks). This comparison reveals

a great overlap in regulated processes and indicates the role of epigenetic modification in CCM hallmark gene expression.

We have included a better explanation of the GO term analysis in the revised version of the manuscript with respect to the RNAseq data and CUT&RUNseq data sets. The description and relevant analysis of this data is also found in the Material and Methods section.

7. Fig. 2U shows upregulation of WNT9B and WNT9A in human CCM lesions and iPSC-derived ECs. It is not clear what the control is here. What is the expression compared to?

The RNASeq data set in revised Fig. 3N was generated from human iPSCs, which were mutagenized using CRISPR/Cas9 gene editing. Three homozygous knock-out clones and three sibling clones that did not show CRISPR-mediated edits were used in the endothelia cell differentiation process. The sibling clones that did not show CRISPR-mediated edits were used as controls. For the differential gene expression analysis, the expression profiles of the CCM2-KO cells were compared to the expression profiles of the control cells. This is described in the materials and methods section. For further clarification, we will add to the figure legend the following sentence:

“(N) Schematic overview of *WNT9A* and *WNT9B* mRNA levels in primary human lesion material from familial and sporadic CCM patients as compared to healthy brain material and in human iPSC-derived endothelial cells depleted of CCM2 as compared to non-edited cells. In all cases, *WNT9A* and *WNT9B* are upregulated.”

Minor points.

1. Please check the reference to Fig. 3U on page 11, line 279; perhaps it should be Fig. 2U.

We have corrected this error.

2. Authors describe in the discussion that zebrafish *cbx7a* mutants were phenotypically normal (Fig. S5). This data should be presented much earlier in the results section, when *cbx7a* mutant generation is first described.

We have moved this description to where the *cbx7a* mutant is first described.

3. Please check the reference to Fig. S5 in the Methods section, page 30, line 784; Fig. S5 does not show karyotyping.

We have corrected this error.

10th Sep 2024

Dear Dr. Seyfried,

Thank you for submitting your revised study. We have now received the feedback from the referees, and as you will see below, they are overall satisfied with the revisions. I am therefore pleased to inform you that I will be able to accept your manuscript once the following minor concerns are addressed:

1/ Referees' comments:

Please address the minor comments from referee #2.

2/ Manuscript text:

- Please accept previous changes and only keep in track changes mode any new modification.
- The emails from authors Meline Hussmann (melina.hussmann@ukmuenster.de) and Benjamin Keith (benjamin.keith@astrazeneca.com) bounced, please check and correct accordingly.
- All corresponding authors are required to supply an ORCID ID for their name upon submission of a revised manuscript. Currently, an ORCID identifier is missing for M. Pagani. Please note that we cannot register ORCID ID on behalf of authors.
- Please reorder the sections of the manuscript as follows: Abstract, Introduction, Results, Discussion, Methods, Data Availability, Acknowledgements, Disclosure and competing interests statement, The Paper Explained, References, Figure legends, Expanded View Figure legends.
- Methods:
 - o Cells: please indicate whether HUVEC cells were tested for mycoplasma contamination
 - o Human samples: please include the sentence that the experiments conformed to the principles set out in the WMA Declaration of Helsinki and the Department of Health and Human Services Belmont Report.
 - o Mice: please indicate the origin of the mice, as well as age and gender at time of experiments.
- Data Availability: primary datasets (i.e. RNAseq) produced in this study need to be deposited in an appropriate public database, and the accession numbers and database should be listed in this section.
- Acknowledgements: funding listed in the submission system and the manuscript should match (currently, DFG grant INST 336/114-1 FUGG, ANRT, Fondazione AIRC under 5 per mille 2019, ID 22759 program, Telethon1238 GGP19202, and Ministero dell'Istruzione, dell'Università e della Ricerca award PNRR1239 M4C2-Investimento 1.4-CN0000041 - NextGenerationEU are not entered in the submission system).

3/ Figures:

- Please make sure that all figures and figure panels are referenced in the text, and in chronological order (currently, Fig 4A is called out before Fig 3O-Y; panel callouts are missing for Fig EV1 and EV3).
- Dataset EV legends: Tables EV1 and EV2 should be made Dataset EV1 and Dataset EV2 and need their legends removed from the manuscript text and added to the corresponding files, in a separate tab/worksheet; Table EV3 should consequently be renamed Table EV1 and it also needs its legend removed from the manuscript and added to the top of the page. Callouts in the manuscript text will need to be updated accordingly.
- Appendix: a table of contents should be added to the first page, with page numbers.
- Please address the queries from our copy editors in the figure legends:
 1. Please note that the exact p values are not provided in the legends of figures 2b, f.
 2. Please indicate the statistical test used for data analysis in the legends of figures 1a; 3a; EV 5a-d.
 3. Please note that the box plots need to be defined in terms of minima, maxima, centre, bounds of box and whiskers, and percentile in the legends of figures EV 2k-m.
 4. Please note that information related to n is missing in the legends of figures 1f; 2b, f-g, l, s; 3c, o-q; EV 2k-m.
 5. Please note that the scale bar is missing for figures 2c'-e'.
 6. Please note that the scale bar needs to be defined for figures 2n-r; 3j-m.
 7. Please note that scale bar and its definition are missing for figures EV 1c'-d''.
 8. Please note that the asterisk and arrows are not defined in the legend of figures 2c'-e'; 3f', i'. This needs to be rectified.
 9. Please note that the red arrowheads are not defined in the legend of figure 3i, u-v; EV 2b, h, n, n', q'; EV 3a. This needs to be rectified.
 10. Please note that the black arrowheads are not defined in the legend of figure EV 4d-f. This needs to be rectified.
 11. Please note that the white arrows/ arrowheads are not defined in the legend of figure EV 1d'-d'', EV 2q"; EV 3a"; EV 4j, k', m'-n'. This needs to be rectified.

4/ Checklist:

- I am not sure restrictions on newly created materials apply to your study, could you please check and clarify?
- please fill in the section Cell materials/authentication & mycoplasma contamination
- please double check that you do not need to fill in the section "Core facilities"
- please fill in all sections of "Experimental study design and statistics"

- please fill in the right column for "Data Availability"/ "Data citations"

5/ I introduced minor changes in your Paper Explained. Please let me know if you agree, or amend as you see fit:

PROBLEM

Cerebral cavernous malformation (CCM) is a vascular disease caused by mutations in the CCM1/KRIT1, CCM2, or CCM3/PDCD10 genes. These mutations trigger the transcription factor Krüppel-like factor 2 (KLF2) to activate pathological changes in endothelial cells gene expression. Identifying the genes that become activated by KLF2 in the pathology of CCM is of great biomedical interest.

RESULTS

Our research revealed that the Chromobox Protein Homolog 7 (CBX7), which modifies genome organization, also changes the genes activated by KLF2. Higher levels of CBX7 were found in endothelial cells of patients with CCM and in mouse and zebrafish pre-clinical CCM disease models, compared with healthy subjects. The pathological CCM phenotypes in zebrafish embryos, CCM2-deficient human umbilical vein endothelial cells (HUVECs), and in a pre-clinical CCM3 mouse disease model, were suppressed by genetic or pharmacological CBX7 inhibition. Furthermore, both KLF2 and blood flow regulated the expression of CBX7 in zebrafish. In turn, CBX7 was involved in the activation of KLF2 target genes, including genes related to endothelial-to-mesenchymal transition (endoMT), WNT9 and TEK/TIE2.

IMPACT

Our findings show that CBX7 directs KLF2 towards pathological target genes. This discovery offers potential therapeutic strategies, as targeting CBX7 may suppress the harmful effects of CCM mutations in patients and improve their condition.

6/ I slightly edited your synopsis, please let me know if you agree with the following, or amend as you see fit:

"A novel therapeutic approach for the treatment of cerebral cavernous malformation (CCM) was established through targeting the polycomb repressive complex 1 protein CBX7.

- In CCM, brain endothelial cells exhibit increased epigenetic modifications due to activity of the polycomb repressive complex 1 protein CBX7.
- Changes to the epigenetic landscape in affected endothelial cells trigger a pathological gene expression that involves TEK, ANGPT1, WNT9, and endoMT genes.
- The activity of CBX7 is regulated by the transcriptional regulator KLF2 and blood flow.
- Genetic ablation or pharmacological inhibition of CBX7 in pre-clinical zebrafish, mouse, and human endothelial cell models suppresses cerebral cavernous malformation phenotypes."

Please resize your visual abstract 550 px wide x 300-600 px high. A cropped portion of this image will serve as thumbnail for the table of content on our webpage.

7/ As part of the EMBO Publications transparent editorial process initiative (see our Editorial at <http://embomolmed.embopress.org/content/2/9/329>), EMBO Molecular Medicine will publish online a Review Process File (RPF) to accompany accepted manuscripts.

This file will be published in conjunction with your paper and will include the anonymous referee reports, your point-by-point response and all pertinent correspondence relating to the manuscript. Let us know whether you agree with the publication of the RPF.

I look forward to receiving your revised manuscript.

With kind regards,

Lise Roth

***** Reviewer's comments *****

Referee #1 (Comments on Novelty/Model System for Author):

This manuscript focuses on the novel role of CBX7 and its potential downstream signaling in CCM2 models, using zebrafish as well as mouse and human samples, and explores potential drug candidates to partially rescue the CCM phenotype. This research could benefit the clinical management of the disease

Referee #1 (Remarks for Author):

After major revisions, this manuscript has significantly improved both logically and technically, incorporating high-quality images, new data and models. The authors have addressed all my concerns and questions, and I have no further comments.

Referee #2 (Remarks for Author):

The manuscript has clearly improved with the inclusion of new data and the reorganization of some of the figures. I only have a couple of minor comments in the text:

1-In line 362, authors should include the name of the gene (WNT9B), instead of saying "this gene". Otherwise, it is not clear, given the paragraph in the middle.

2-In line 520 and 36, PIK3CA is misspelled if they refer to the gene (should be *PIK3CA* in italics). Yet, if they refer to the protein, they should replace it for PI3Ka or p110a, as these are the correct ways to refer to the protein.

Referee #3 (Comments on Novelty/Model System for Author):

The manuscript by Pham et al describes the role of CBX7, a component of the Polycomb Repressive Complex1, during CCM signaling in zebrafish embryos and human endothelial cells. The authors demonstrated that Cbx7 homolog expression is upregulated in CCM-deficient zebrafish embryos and human cells, and that mutation or inhibition of Cbx7 function reverses CCM phenotypes. They utilize transcriptomic approaches to demonstrate that Cbx7 plays a role in activation of Klf2 targeting genes including Wnt9 and others. Altogether, this study reveals a mechanistic role for CBX7 during CCM pathogenesis. Overall these results are novel and highly significant, as they contribute to mechanistic understanding of CCM pathogenesis. Experiments are well performed, and results are clear and strongly support conclusions.

Referee #3 (Remarks for Author):

In the revised manuscript the authors have appropriately addressed all of my previous concerns. I do not have any other concerns.

The authors have addressed all minor editorial requests.

27th Sep 2024

Dear Salim,

Thank you for submitting the revised files. I am pleased to inform you that your manuscript is accepted for publication and is now being sent to our publisher to be included in the next available issue of EMBO Molecular Medicine.

With kind regards,

Lise
